# SHARPER BOUNDS OF NON-CONVEX STOCHASTIC GRADIENT DESCENT WITH MOMENTUM

## ABSTRACT

Stochastic gradient descent with momentum (SGDM) has been widely used in machine learning. However, in non-convex domains, high probability learning bounds for SGDM are scarce. In this paper, we provide high probability convergence bounds and generalization bounds for SGDM. Firstly, we establish these bounds for the gradient norm in the general non-convex case. The derived convergence bounds are tighter than the theoretical results of related work, and to our best knowledge, the derived generalization bounds are the first ones for SGDM. Then, if the Polyak-Łojasiewicz condition is satisfied, we establish these bounds for the error of the function value, instead of the gradient norm. Moreover, the derived learning bounds have faster rates than the general non-convex case. Finally, we further provide sharper generalization bounds by considering a mild Bernstein condition on the gradient. In the case of low noise, their learning rates can reach $\widetilde{\mathcal{O}}(1/n^2)$, where $n$ is the sample size. Overall, we relatively systematically investigate the high probability learning bounds for non-convex SGDM.

## 1 INTRODUCTION

Stochastic optimization plays an essential role in modern statistical and machine learning, as many machine learning problems can be cast into stochastic optimization problems. The last decades have seen much significant progress in the development of stochastic optimization algorithms, of which stochastic gradient descent with momentum (SGDM) has drawn a lot of attention on a broad range of problems due to its simplicity and its low computational complexity per update (Goodfellow et al., 2016; Li & Orabona, 2020). As a fundamental algorithm for stochastic optimization, SGDM has shown tremendous success in natural language understanding, computer vision, and speech recognition (Krizhevsky et al., 2012; Hinton et al., 2012; Sutskever et al., 2013). Particularly, SGDM has been widely used to accelerate the back-propagation algorithm in the training of deep neural networks (Rumelhart et al., 1986; Sutskever et al., 2013). Typically, SGDM adds a momentum term to stochastic gradient descent (SGD) in updating the solution, i.e, the difference between the current iterate and the previous iterate. The intuition behind SGDM is that if the direction from the previous iterate to the current iterate is "correct", SGDM utilizes this inertia weighted by the momentum parameter, instead of just relying on the current point used in SGD. Much of the state-of-the-art empirical performance has been achieved with SGDM (Huang et al., 2017; Howard et al., 2017; He et al., 2016; Kim et al., 2021a). Yet, from a theoretical point of view, the analysis of the learning bounds of SGDM is not sufficiently well-documented (Li et al., 2022; Li & Orabona, 2020).

The learning bound of SGDM can be studied from two perspectives: the convergence bound and the generalization bound. The former focuses on how the learning algorithm optimizes the empirical risk, and the latter concerns how the learned model from training samples performs on the testing points (Lei et al., 2021b). From the perspective of the convergence bound, existing literature of convergence for SGDM or deterministic gradient descent with momentum (DGDM) in the non-convex domain mostly uses an analysis of expectation (Ochs et al., 2014; 2015; Ghadimi et al., 2015; Lessard et al., 2016; Yang et al., 2016; Wilson et al., 2021; Gadat et al., 2018; Orvieto et al., 2020; Can et al., 2019; Li et al., 2022; Yan et al., 2018; Liu et al., 2020), to mention but a few. However, the expected bound does not rule out extremely bad outcomes (Li & Orabona, 2020; Liu et al., 2023). Moreover, in practical applications such as machine learning, it is often the case that the algorithm is usually run only once since the training process may take a long time. Therefore, high probability

bounds, as compared to expectation bounds, are preferred in the study of the performance of the algorithm on single runs (Harvey et al., 2019). To our best knowledge, there are only two works on the high probability convergence bound of SGDM (Li & Orabona, 2020; Cutkosky & Mehta, 2021). Specifically, Cutkosky & Mehta (2021) consider the case that the gradient follows a $\theta$-order moment condition, $\theta \in (1, 2]$, and presents a $\widetilde{\mathcal{O}}(T^{-\frac{\theta-1}{3\theta-2}})$ convergence rate for the gradient norm, where $T$ is the iterate number. Li & Orabona (2020) provide a convergence bound of the order $\widetilde{\mathcal{O}}(1/\sqrt{T})$ for the square gradient norm by considering the sub-Gaussian gradient noise. It is then discussed in (Li et al., 2022) that it is unclear if this convergence rate can be improved and can be extended to more general settings beyond the sub-Gaussian gradient noise. In general, these convergence bounds in (Li & Orabona, 2020; Cutkosky & Mehta, 2021) are of the slow order, and there are no generalization bounds are given in (Li & Orabona, 2020; Cutkosky & Mehta, 2021).

From the perspective of the generalization bound, existing generalization studies of SGDM and DGDM are scarce. Ong (2017); Chen et al. (2018) provide expected generalization error bounds for a specific quadratic loss function of DGDM by the lens of algorithmic stability (Bousquet & Elisseeff, 2002; Hardt et al., 2016). The analysis of (Ong, 2017; Chen et al., 2018) cannot be easily extended to general loss functions. It is conjectured in (Chen et al., 2018) that the uniform stability bound they derived may also be applicable to general convex loss functions. Motivated by this, Ramezani-Kebrya et al. (2024) study the generalization error bound of SGDM for the general loss functions. Surprisingly, however, their analysis shows a counterexample for which the uniform stability gap (in expectation, i.e., taking expectation over the internal randomness of the learning algorithm) for SGDM running multiple epochs diverges even for the convex loss functions. It is also revealed in (Attia & Koren, 2021) that in the general convex case, the uniform stability gap of the deterministic Nesterov's accelerated gradient algorithm (NAG) collapses exponentially fast with the number of iterates. We remind the readers here that the uniform stability is only a sufficient condition for generalization, and it is unclear how other weaker stability measures, such as on-average stability (Shalev-Shwartz et al., 2010), would behave on the generalization analysis of SGDM. Overall, there are known difficulties in developing generalization performance guarantees of SGDM, especially for general loss functions. Furthermore, similar to the analysis of SGDM's convergence, high probability generalization bounds are also more challenging to derive compared to bounds on expectations (Bousquet et al., 2020; Bassily et al., 2020; Feldman & Vondrak, 2019; Li & Liu, 2022).

Therefore, both the high probability convergence bound and generalization bound of SGDM are far from being understood. Motivated by the problems we discussed above, this paper makes an attempt to establish high probability convergence bounds and generalization bounds for SGDM, particularly, in non-convex settings. For brevity, from now on, all bounds on the performance of the learned model on testing data, such as the generalization error bound and the excess risk bound, will be called generalization bounds. Our contributions can be summarized as follows:

1) On a high level, we study the case where the stochastic gradient noise follows a novel class of sub-Weibull distribution (Vladimirova et al., 2019; 2020; Kuchibhotla & Chakrabortty, 2018), which generalizes the sub-Gaussian noise considered in (Li & Orabona, 2020) to potentially heavier-tailed ones. Our learning bounds under this distribution can show the impact of moving from sub-Gaussian/sub-exponential (i.e. light-tailed) variables to those with heavy exponential tails on the rates of convergence and generalization.

2) We first provide a high probability analysis for SGDM in the general non-convex case. In this case, we establish convergence bounds of the order $\widetilde{\mathcal{O}}(1/T^{\frac{1}{2}})$ and generalization bounds of the order $\widetilde{\mathcal{O}}(d^{\frac{1}{2}}/n^{\frac{1}{2}})$ for the square gradient norm, where $d$ is the dimension and $n$ is the sample size. The convergence bounds are tighter than those of the related work. Moreover, to our best knowledge, the high probability generalization bounds are the first ones for SGDM.

3) We then perform a high probability analysis for SGDM with Polyak-Łojasiewicz property on non-convex objectives. In this case, we establish sharper convergence bounds of the order $\widetilde{\mathcal{O}}(1/T)$. Furthermore, the bounds are derived for the last iterate of SGDM and the error of the function value, instead of the average iterate and the gradient norm studied in the general non-convex case. Moreover, we provide generalization bounds of the faster order $\widetilde{\mathcal{O}}\left(\frac{d+\log(\frac{1}{\delta})}{n}\right)$ for SGDM, which have never been given before.

4) We finally consider a mild Bernstein condition on the gradient. In this case, we improve the $\widetilde{\mathcal{O}}\big(\frac{d+\log(\frac{1}{\delta})}{n}\big)$ order generalization bound to the order $\widetilde{\mathcal{O}}(1/n^2 + F^*/n)$, where $F^*$ is the optimal population risk. In the low noise case where $F^*$ is tiny, this bound allows a faster $\widetilde{\mathcal{O}}(1/n^2)$ learning rate, which shows a tighter dependency on the sample size $n$. Another positive point of this bound is that we successfully remove the dimension parameter $d$, allowing it to easily incorporate massive neural networks that are often high-dimensional.

In conclusion, by considering different conditions on the objective functions, we successfully establish improved learning bounds with different rates, which systematically demonstrates the learning guarantee of SGDM from the perspective of convergence and generalization. The paper is organized as follows. The preliminaries are given in Section 2. The main results are provided in Section 3. Numerical experiments are then reported in Section 4. We conclude this paper in Section 5. The proofs are postponed to the Appendix.

## 2 PRELIMINARIES

### 2.1 NOTATIONS

Let $\mathcal{X}$ be a parameter space in $\mathbb{R}^d$ and $\mathbb{P}$ be a probability measure defined on a sample space $\mathcal{Z}$. Denote $f : \mathcal{X} \times \mathcal{Z} \mapsto \mathbb{R}_+$. We consider the following stochastic optimization algorithm

$$\min_{\mathbf{x} \in \mathcal{X}} F(\mathbf{x}) := \mathbb{E}_{z \sim \mathbb{P}}[f(\mathbf{x}; z)],$$

where $F$ is often referred to as population risk, $f$ is possible non-convex, and $\mathbb{E}_{z \sim \mathbb{P}}$ denotes the expectation with respect to (w.r.t.) the random variable $z$ drawn from $\mathbb{P}$. In practice, $\mathbb{P}$ is unavailable and what we get is a dataset $S = \{z_1, ..., z_n\}$ independently and identically drawn from the underlying $\mathbb{P}$. One typically instead optimize the following empirical risk

$$\min_{\mathbf{x} \in \mathcal{X}} F_S(\mathbf{x}) := \frac{1}{n} \sum_{i=1}^{n} f(\mathbf{x}; z_i).$$

To optimize the $F_S(\mathbf{x})$, SGDM have been widely adopted (Polyak, 1964; Qian, 1999; Sutskever et al., 2013; Li & Orabona, 2020). In this work we focus on Polyak's momentum, also known as the Heavy-ball algorithm or classic momentum, which is arguably the most popular form of momentum in current machine learning practice (Liu et al., 2020). The pseudocodes of SGDM (Polyak's momentum) are shown in Algorithm 1. Vanilla SGD's update is $\mathbf{x}_{t+1} = \mathbf{x}_t - \eta_t \nabla f(\mathbf{x}_t; z_{j_t})$. In step 3 of Algorithm 1, SGDM adds a momentum term $\mathbf{m}_{t-1}$ weighted by a momentum parameter $\gamma$ to the gradient estimate $\nabla f(\mathbf{x}_t; z_{j_t})$ of SGD. And in step 4, SGDM updates the solution with $\mathbf{x}_{t+1} = \mathbf{x}_t - \mathbf{m}_t$. Thus, SGDM's update is $\mathbf{x}_{t+1} = \mathbf{x}_t - \eta_t \nabla f(\mathbf{x}_t; z_{j_t}) + \gamma(\mathbf{x}_t - \mathbf{x}_{t-1})$.

Let us introduce some notations to simplify the presentation. Let $B = \sup_{z \in \mathcal{Z}} \|\nabla f(\mathbf{0}; z)\|$, where $\nabla f(\cdot; z)$ denotes the gradient of $f$ w.r.t. the first argument and $\|\cdot\|$ denotes the Euclidean norm. For any $R > 0$, we define $B(\mathbf{x}_0, R) := \{\mathbf{x} \in \mathbb{R}^d : \|\mathbf{x} - \mathbf{x}_0\| \le R\}$ which denotes a ball with center $\mathbf{x}_0 \in \mathbb{R}^d$ and radius $R$. Let $\mathbf{x}(S) \in \arg\min_{\mathbf{x} \in \mathcal{X}} F_S(\mathbf{x})$ and $\mathbf{x}^* \in \arg\min_{\mathcal{X}} F(\mathbf{x})$. Denoted by $a \asymp b$ if there exists universal constants $c, c' > 0$ such that $ca \le b \le c'a$. In this paper, the standard order of magnitude notations such as $\mathcal{O}(\cdot)$ and $\widetilde{\mathcal{O}}(\cdot)$ will be used.

### 2.2 ASSUMPTIONS

We need some assumptions. The following assumptions are scattered in different Theorems.

**Assumption 2.1.** The differentiable function $f$ is a (possibly) non-convex function and for any $z \in \mathcal{Z}$, $\mathbf{x} \mapsto f(\mathbf{x}; z)$ is $L$-smooth. A differentiable function $g : \mathcal{X} \mapsto \mathbb{R}$ is called $L$-smooth with $L > 0$ if the following inequality holds for every $\mathbf{x}_1, \mathbf{x}_2$:

$$\|\nabla g(\mathbf{x}_1) - \nabla g(\mathbf{x}_2)\| \le L\|\mathbf{x}_1 - \mathbf{x}_2\|,$$

where $\nabla$ is the gradient operator. More properties on the smothness are shown in Lemma B.7.

**Assumption 2.2.** The gradient at $\mathbf{x}^*$ satisfies the Bernstein condition: there exists $B_* > 0$ such that for all $2 \le k \le n$,

$$\mathbb{E}_z \left[\|\nabla f(\mathbf{x}^*; z)\|^k\right] \le \frac{1}{2}k!\mathbb{E}_z \left[\|\nabla f(\mathbf{x}^*; z)\|^2\right] B_*^{k-2}.$$

*Remark* 2.3. The Bernstein condition is common in learning theory. It was shown in (Wainwright, 2019) that given a random variable $X$ with mean $\mu = \mathbb{E}[X]$ and variance $\sigma^2 = \mathbb{E}[X^2] - \mu^2$, we say that Bernstein's condition with parameter $b$ holds if for $k = 2, ...$, there holds $\mathbb{E}\left[(X - \mu)^k\right] \leq \frac{1}{2}k!\sigma^2 b^{k-2}$. In fact, the Bernstein condition is nearly equivalent to being sub-exponential, refer to a discussion in Remark 4 in (Lei, 2020). The classical sub-Gaussian and sub-exponential distributions all satisfy this condition. For these distributions, their $k$-order moments are bounded by the second-order moment. In other words, the Bernstein condition is mild, for example, weaker than the bounded assumption of random variables. Thus, Assumption 2.2 is a Bernstein condition on the variable $\|\nabla f(\mathbf{x}^*; z)\|$, which is weaker than that $\|\nabla f(\mathbf{x}^*; z)\|$ is bounded, while the latter, bounded gradient norm condition, is widely used in stochastic optimization (Zhang et al., 2017).

**Assumption 2.4.** For all $S \in \mathcal{Z}^n$, and for some positive $G$, we have

$$\eta_t \|\nabla F_S(\mathbf{x}_t)\| \leq G, \quad \forall t \in \mathbb{N}.$$

*Remark* 2.5. In the literature of theoretical analysis of SGDM, a bounded gradient assumption as $\|\nabla f(\mathbf{x}_t; z)\| \leq G$, also referred to as the Lipschitz continuity of $f$ (Lei et al., 2019), is standard (Li et al., 2022; Li & Orabona, 2020; Li & Liu, 2023). Assumption 2.4 relaxes the bounded gradient assumption by multiplying the stepsize $\eta_t$ and replacing $\nabla f(\mathbf{x}_t; z)$ with $\nabla F_S(\mathbf{x}_t)$. The stepsize $\eta_t$ would decrease to zero for the convergence of the algorithm. Moreover, typical decay rates of the stepsize $\eta_t$ are $\mathcal{O}(t^{-\frac{1}{2}})$ and $\mathcal{O}(t^{-1})$ (Lei & Tang, 2021), in which case, the gradients of $F_S$ can respectively grow with the rate $\mathcal{O}(t^{\frac{1}{2}})$ and $\mathcal{O}(t)$ without violating this assumption.

In the next, we introduce the Polyak-Łojasiewicz condition.

**Assumption 2.6.** Fix a set $\mathcal{X}$ and let $f^* = \min_{\mathbf{x} \in \mathcal{X}} f(\mathbf{x})$. For any function $f : \mathcal{X} \mapsto \mathbb{R}$, we say it satisfies the Polyak-Łojasiewicz condition with parameter $\mu > 0$ on $\mathcal{X}$ if for all $\mathbf{x} \in \mathcal{X}$,

$$f(\mathbf{x}) - f^* \leq \frac{1}{2\mu}\|\nabla f(\mathbf{x})\|^2.$$

*Remark* 2.7. Fast rates cannot be achieved for free. The Polyak-Łojasiewicz condition is widely used in the optimization community to obtain fast convergence rates (Necoara et al., 2019; Karimi et al., 2016) and is one of the weakest curvature conditions to replace the strong convexity (Karimi et al., 2016). This condition can be viewed as a specific instance of the Kurdyka-Łojasiewicz condition. The Kurdyka-Łojasiewicz condition is prevalent, as it has been shown that all analytic and semi-algebraic functions satisfy such a condition (Bolte et al., 2010; Attouch & Bolte, 2009; Attouch et al., 2010; Bolte et al., 2014).

In the sequel, we make an assumption on the noise of the stochastic gradient.

**Assumption 2.8.** The gradient noise $\nabla f(\mathbf{x}_t; z_{j_t}) - \nabla F_S(\mathbf{x}_t)$ satisfies

$$\mathbb{E}_{j_t}\left[\exp(\|\nabla f(\mathbf{x}_t; z_{j_t}) - \nabla F_S(\mathbf{x}_t)\|/K)^{\frac{1}{\theta}}\right] \leq 2, \tag{1}$$

for some positive $K$ and $\theta \geq 1/2$.

*Remark* 2.9. Li & Orabona (2020) assume $\mathbb{E}_{j_t}\left[\exp(\|\nabla f(\mathbf{x}_t; z_{j_t}) - \nabla F_S(\mathbf{x}_t)\|^2/K^2)\right] \leq 2$, which implies that the tails of the noise distribution are dominated by tails of a Gaussian distribution. As a comparison, Assumption 2.8 generalizes the sub-Gaussian noise to a richer class of distributions, including the sub-Exponential distribution (i.e., $\theta = 1$) and heavier-tailed distributions (i.e., $\theta > 1$). Indeed, the distributions in (1) is called the sub-Weibull distribution (Vladimirova et al., 2020): a random variable $X$, satisfying $\mathbb{E}\left[\exp\left((|X|/K)^{\frac{1}{\theta}}\right)\right] \leq 2$, for some positive $K$ and $\theta$, is called a sub-Weibull random variable with tail parameter $\theta$. The higher tail parameter $\theta$ corresponds to the heavier tails (Kuchibhotla & Chakrabortty, 2018). Thus, the learning bounds in this paper hold for a broad class of heavy-tailed distributions. Our motivation for studying the heavy-tailed sub-Weibull noise of stochastic gradients is that it indicates the impact of moving from sub-Gaussian/sub-exponential (i.e. light-tailed) variables to those with heavy exponential tails on the rates of convergence and generalization and that many recent works suggest that stochastic optimization algorithms have heavier noise than sub-Gaussian (Panigrahi et al., 2019; Madden et al., 2021; Gurbuzbalaban et al., 2021; Simsekli et al., 2019; Şimşekli et al., 2019; Zhang et al., 2020; 2019; Wang et al., 2021; Gurbuzbalaban & Hu, 2021).

---

**Algorithm 1** SGD with Momentum (SGDM)

---

**Require:** stepsizes $\{\eta_t\}_t$, dataset $S = \{z_1, ..., z_n\}$, and momentum parameter $0 < \gamma < 1$.
**Initializtion:** $\mathbf{x}_1 = \mathbf{0}, \mathbf{m}_0 = \mathbf{0}$,

1: **for** $t = 1, ..., T$ **do**
2:     sample $j_t$ from the uniform distribution over the set $\{j : j \in [n]\}$,
3:     update $\mathbf{m}_t = \gamma \mathbf{m}_{t-1} + \eta_t \nabla f(\mathbf{x}_t; z_{j_t})$,
4:     update $\mathbf{x}_{t+1} = \mathbf{x}_t - \mathbf{m}_t$.
5: **end for**

---

## 3 MAIN RESULTS

This section introduces our main theoretical results.

### 3.1 LEARNING BOUNDS IN GENERAL NON-CONVEX CASE

In the general noncovex case, we are interested in finding a first-order $\epsilon$-stationary point satisfying $\|\nabla F_S(\mathbf{x}_t)\|^2 \leq \epsilon$ for the convergence bound and $\|\nabla F(\mathbf{x}_t)\|^2 \leq \epsilon$ for the generalization bound, since in this case we cannot guarantee that the algorithm can find a global minimizer. As the standard measure in the general non-convex case, we will quantify the optimization performance and generalization performance w.r.t. the average square gradient norm $\frac{1}{T}\sum_{t=1}^{T} \|\nabla F_S(\mathbf{x}_t)\|^2$ and $\frac{1}{T}\sum_{t=1}^{T} \|\nabla F(\mathbf{x}_t)\|^2$, respectively.

#### 3.1.1 CONVERGENCE BOUNDS

We first provide convergence bounds with high probabilities for SGDM. The convergence bound characterizes how the optimization algorithm minimizes the empirical risk $F_S$.

**Theorem 3.1.** *Let $\mathbf{x}_t$ be the sequence of iterates generated by Algorithm 1. Set the stepsize as $\eta_t = ct^{-\frac{1}{2}}$, where $c \leq \frac{1}{4} \frac{(1-\gamma)^3}{3L - L\gamma}$.*

*(1.) If $\theta = \frac{1}{2}$, we suppose Assumptions 2.1 and 2.8 hold. For any $\delta \in (0, 1)$, with probability $1 - \delta$, we have the following inequality*

$$\frac{1}{T} \sum_{t=1}^{T} \|\nabla F_S(\mathbf{x}_t)\|^2 = \mathcal{O}\Big(\frac{\log(1/\delta) \log T}{\sqrt{T}}\Big).$$

*(2.) If $\frac{1}{2} < \theta \leq 1$, we suppose Assumptions 2.1, 2.4, and 2.8 hold. For any $\delta \in (0, 1)$, with probability $1 - \delta$, we have the following inequality*

$$\frac{1}{T} \sum_{t=1}^{T} \|\nabla F_S(\mathbf{x}_t)\|^2 = \mathcal{O}\Big(\frac{\log^{2\theta}(1/\delta) \log T}{\sqrt{T}}\Big).$$

*(3.) If $\theta > 1$, we suppose Assumptions 2.1, 2.4, and 2.8 hold. For any $\delta \in (0, 1)$, with probability $1 - \delta$, we have the following inequality*

$$\frac{1}{T} \sum_{t=1}^{T} \|\nabla F_S(\mathbf{x}_t)\|^2 = \mathcal{O}\Big(\frac{\log^{\theta-1}(T/\delta) \log(1/\delta) + \log^{2\theta}(1/\delta) \log T}{\sqrt{T}}\Big).$$

*Remark* 3.2. The convergence bounds established here are of the order $\widetilde{\mathcal{O}}(1/\sqrt{T})$. Theorem 3.1 reveals that bigger $\theta$ gives convergence bounds with slower rates, which confirms the intuition that heavier-tailed gradient noise, i.e., bigger $\theta$, results in worse convergence. We compare these bounds with the related work (Li & Orabona, 2020; Cutkosky & Mehta, 2021). Cutkosky & Mehta (2021) study a different setting which is a combination of gradient clipping, momentum (not Polyak's momentum) and normalized gradient descent. Their Theorem 2 provides a convergence bound of the order $\mathcal{O}\Big(\frac{\log(T/\delta)}{T^{\frac{\theta-1}{3\theta-2}}}\Big)$ for $\frac{1}{T}\sum_{t=1}^{T} \|\nabla F_S(\mathbf{x}_t)\|$ under the smoothness condition and a $\theta$-order moment

condition of the gradient, where $\theta \in (1,2]$. In the case of $\theta = 2$, this bound achieves $\widetilde{\mathcal{O}}\left(\frac{1}{T^{1/4}}\right)$ rate. According to the Jensen's inequality, we have $(\frac{1}{T}\sum_{t=1}^{T}\|\nabla F_S(\mathbf{x}_t)\|)^2 \leq \frac{1}{T}\sum_{t=1}^{T}\|\nabla F_S(\mathbf{x}_t)\|^2$. Thus, the convergence bounds in Theorem 3.1 imply $\frac{1}{T}\sum_{t=1}^{T}\|\nabla F_S(\mathbf{x}_t)\| = \widetilde{\mathcal{O}}\left(\frac{1}{T^{1/4}}\right)$. It has recently been shown that the expected $\mathcal{O}(1/T^{1/4})$ rate is optimal in the worst case (Arjevani et al., 2019). Li & Orabona (2020) study Polyak's momentum and their Theorem 1 provides a convergence bound of the order $\mathcal{O}\left(\frac{\log(T/\delta)\log T}{\sqrt{T}}\right)$ for $\frac{1}{T}\sum_{t=1}^{T}\|\nabla F_S(\mathbf{x}_t)\|^2$ under the smoothness condition and the specific case $\theta = 1/2$. Theorem 3.1 slightly refines this bound to $\mathcal{O}\left(\frac{\log(1/\delta)\log T}{\sqrt{T}}\right)$ under the same conditions. Although this improvement is marginal, other bounds of Theorem 3.1 that generalize the sub-Gaussian case to heavier-tailed distributions are novel. Theorem 2 in (Li & Orabona, 2020) also studies a variant of AdaGrad with Polyak's momentum, called delayed AdaGrad whose stepsize doesn't contain the current gradient (Li & Orabona, 2019). The convergence bound of delayed AdaGrad established in (Li & Orabona, 2020) is of the order $\max\left\{\mathcal{O}\left(\frac{d\log^{\frac{3}{2}}(T/\delta)}{\sqrt{T}}\right), \mathcal{O}\left(\frac{d^2\log^2(T/\delta)}{T}\right)\right\}$. When dimension $d$ is small, this bound shows a rate of the order $\mathcal{O}\left(\frac{d\log^{\frac{3}{2}}(T/\delta)}{\sqrt{T}}\right)$. As a comparison, convergence bounds in Theorem 3.1 are clearly sharper. Note that this work studies Polyak's momentum, so the results of (Li & Orabona, 2020) are more comparable to ours.

There are many applications that validate the empirical advantages of SGDM compared with SGD. Although the bound of Theorem 3.1 is optimal w.r.t. $T$, our results fail to explain the advantage of SGDM over SGD. In the convex setting, there have been relevant results proving SGDM's superiority over SGD [1]. However, in the non-convex setting, how to provide a bound that can explicitly demonstrate that SGDM is still better than SGD is a known challenge, and this viewpoint has been commonly elaborated in existing analysis, for example, the results for algorithms with Polyak's momentum (Li & Orabona, 2020; Zou et al., 2018) and many analyses of Adam and their variants (Luo et al., 2018; Liu et al., 2019; Shi et al., 2021; Chen et al., 2019; Zaheer et al., 2018). In addition, since our work considers the high probability bound, which implies the bound must hold for even the worst-case value of the sample space, this strict requirement may make it more difficult to analyze the advantages of SGDM over SGD. The main purpose of this work is to provide sharper bounds than the existing results of SGDM. It would be our future work to strictly prove that SGDM is better than SGD in non-convex learning.

### 3.1.2 GENERALIZATION BOUNDS

We then provide high probability generalization bounds for SGDM. Generalization characterizes how the learned models from training samples perform on the underlying distribution.

**Theorem 3.3.** *Let $\mathbf{x}_t$ be the sequence of iterates generated by Algorithm 1. Set the stepsize as $\eta_t = ct^{-\frac{1}{2}}$, where $c \leq \frac{1}{4}\frac{(1-\gamma)^3}{3L-L\gamma}$. We choose $T \asymp \frac{n}{d}$.*

*(1.) If $\theta = \frac{1}{2}$, we suppose Assumptions 2.1 and 2.8 hold. For any $\delta \in (0,1)$, with probability $1 - \delta$, we have the following inequality*

$$\frac{1}{T}\sum_{t=1}^{T}\|\nabla F(\mathbf{x}_t)\|^2 = \mathcal{O}\left(\left(\frac{d}{n}\right)^{\frac{1}{2}}\log(\frac{n}{d})\log^3(\frac{1}{\delta})\right).$$

*(2.) If $\frac{1}{2} < \theta \leq 1$, we suppose Assumptions 2.1, 2.4, and 2.8 hold. For any $\delta \in (0,1)$, with probability $1 - \delta$, we have the following inequality*

$$\frac{1}{T}\sum_{t=1}^{T}\|\nabla F(\mathbf{x}_t)\|^2 = \mathcal{O}\left(\left(\frac{d}{n}\right)^{\frac{1}{2}}\log(\frac{n}{d})\log^{(2\theta+2)}(\frac{1}{\delta})\right).$$

*(3.) If $\theta > 1$, we suppose Assumptions 2.1, 2.4, and 2.8 hold. For any $\delta \in (0,1)$, with probability $1 - \delta$, we have the following inequality*

$$\frac{1}{T}\sum_{t=1}^{T}\|\nabla F(\mathbf{x}_t)\|^2 = \mathcal{O}\left(\left(\frac{d}{n}\right)^{\frac{1}{2}}\left(\log(\frac{n}{d})\log^{(2\theta+2)}(\frac{1}{\delta}) + \log^{\theta-1}(\frac{n}{d\delta})\log^2(\frac{1}{\delta})\right)\right).$$

*Remark* 3.4. The generalization bounds provided in Theorem 3.3 are of the order $\widetilde{\mathcal{O}}\big((\frac{d}{n})^{\frac{1}{2}}\big)$. Clearly, bigger $\theta$ gives a slower generalization bound. Similar to Theorem 3.1, Theorem 3.3 suggests that we no longer need Assumption 2.4 when $\theta = 1/2$. To our best knowledge, these generalization bounds are the first ones for SGDM. As discussed in the introduction, the uniform stability tool seems to fail to establish generalization bounds for SGDM with general loss functions. This may be explained as follows: the trade-off between convergence and stability of the algorithm implies that a faster converging algorithm has to be less stable, and vice versa (Chen et al., 2018). Our proof techniques to prove the generalization bounds in this paper belong to the class of the uniform convergence approach (Bartlett & Mendelson, 2002; Bartlett et al., 2005; Xu & Zeevi, 2020; Xu & Zeevi, 2020; Mei et al., 2018; Foster et al., 2018; Davis & Drusvyatskiy, 2021). The uniform convergence can be characterized as that the empirical risk of hypotheses in the hypothesis class converges to their population risk uniformly (Shalev-Shwartz et al., 2010). In the general non-convex case, the dependence of the bound proved by this approach on the dimension $d$ is generally unavoidable (Feldman, 2016), see the results in Theorem 3.3. We highlight here that in Section 3.3 we will successfully remove the dimension $d$ from the generalization upper bound.

## 3.2 LEARNING BOUNDS WITH POLYAK-ŁOJASIEWICZ CONDITION

In the non-convex optimization with the Polyak-Łojasiewicz condition, we are interested in giving upper bounds for the error of the function value. We will quantify the optimization performance and generalization performance w.r.t. $F_S(\mathbf{x}_{T+1}) - F_S(\mathbf{x}(S))$ and $F(\mathbf{x}_{T+1}) - F(\mathbf{x}^*)$, respectively.

### 3.2.1 CONVERGENCE BOUNDS

We first present convergence bounds with high probabilities for SGDM under the Polyak-Łojasiewicz condition.

**Theorem 3.5.** *Let* $\mathbf{x}_t$ *be the sequence of iterates generated by Algorithm 1. Set the stepsize as* $\eta_t = \frac{1}{\mu(S)(t+t_0)}$ *such that* $t_0 \geq \max\{\frac{12L-4L\gamma}{\mu(S)(1-\gamma)^3}, \frac{(8C_\gamma)L}{(1-\gamma)^2\mu(S)} + 1, \frac{8C_\gamma(L\gamma+L\gamma(C_\gamma))}{(1-\gamma)\mu(S)} - 1, 1\}$, *where* $C_\gamma$ *is a constant that depends only on* $\gamma$.

*(1.) If* $\theta = \frac{1}{2}$, *we suppose Assumptions 2.1 and 2.8 hold and suppose the* $F_S$ *satisfies Assumption 2.6 with parameter* $2\mu(S)$. *For any* $\delta \in (0, 1)$, *with probability* $1 - \delta$, *we have the following inequality*

$$F_S(\mathbf{x}_{T+1}) - F_S(\mathbf{x}(S)) = \mathcal{O}\Big(\frac{\log(1/\delta)}{T}\Big).$$

*(2.) If* $\frac{1}{2} < \theta \leq 1$, *we suppose Assumptions 2.1, 2.4 and 2.8 hold and suppose the* $F_S$ *satisfies Assumption 2.6 with parameter* $2\mu(S)$. *For any* $\delta \in (0, 1)$, *with probability* $1 - \delta$, *we have the following inequality*

$$F_S(\mathbf{x}_{T+1}) - F_S(\mathbf{x}(S)) = \mathcal{O}\Big(\frac{\log^{(\theta+\frac{3}{2})}(\frac{1}{\delta})\log^{\frac{1}{2}} T}{T}\Big).$$

*(3.) If* $\theta > 1$, *we suppose Assumptions 2.1, 2.4 and 2.8 hold and suppose the* $F_S$ *satisfies Assumption 2.6 with parameter* $2\mu(S)$. *For any* $\delta \in (0, 1)$, *with probability* $1 - \delta$, *we have the following inequality*

$$F_S(\mathbf{x}_{T+1}) - F_S(\mathbf{x}(S)) = \mathcal{O}\Big(\frac{\log^{(\theta+\frac{3}{2})}(\frac{1}{\delta})\log^{\frac{3(\theta-1)}{2}}(T/\delta)\log^{\frac{1}{2}} T}{T}\Big).$$

*Remark* 3.6. Theorem 3.5 suggests that if the Polyak-Łojasiewicz condition is satisfied, the convergence bounds of SGDM can show fast rates. To be specific, Theorem 3.5 improves the $\widetilde{\mathcal{O}}(1/\sqrt{T})$ rate in Theorem 3.1 to faster $\widetilde{\mathcal{O}}(1/T)$ rate. According to the smoothness property in Remark B.7, we have $\|\nabla F_S(\mathbf{x}_{T+1})\|^2 \leq (2L)(F_S(\mathbf{x}_{T+1}) - F_S(\mathbf{x}(S)))$. Thus, the upper bounds in Theorem 3.5 also hold for the square gradient norm $\|\nabla F_S(\mathbf{x}_{T+1})\|^2$. Moreover, Assumption 2.4 is not required when $\theta = \frac{1}{2}$. Theorem 3.5 also confirms that as $\theta$ increases, the convergence bound gets worse. One can verify easily that these convergence bounds are sharper than the theoretical results of related work (Li & Orabona, 2020; Cutkosky & Mehta, 2021), see Table 1 for details. Also, for the fast $\widetilde{\mathcal{O}}(1/T)$ rate of SGDM in the nonconvex domain, we have not found related results in the literature.

### 3.2.2 GENERALIZATION BOUNDS

We then present high probability generalization bounds for SGDM under the Polyak-Łojasiewicz condition.

**Theorem 3.7.** *Let $\mathbf{x}_t$ be the sequence of iterates generated by Algorithm 1. Set the stepsize as $\eta_t = \frac{1}{\mu(S)(t+t_0)}$ such that $t_0 \geq \max\{\frac{12L-4L\gamma}{\mu(S)(1-\gamma)^3}, \frac{(8C_\gamma)L}{(1-\gamma)^2\mu(S)} + 1, \frac{8C_\gamma(L\gamma+L\gamma(C_\gamma))}{(1-\gamma)\mu(S)} - 1, 1\}$, where $C_\gamma$ is a constant that depends only on $\gamma$. We choose $T \asymp n$.*

*(1.) If $\theta = \frac{1}{2}$, we suppose Assumptions 2.1 and 2.8 hold, assume the $F_S$ satisfies Assumption 2.6 with parameter $2\mu(S)$, and suppose the $F$ satisfies Assumption 2.6 with parameter $2\mu$. For any $\delta \in (0,1)$, with probability $1 - \delta$, we have the following inequality*

$$F(\mathbf{x}_{T+1}) - F(\mathbf{x}^*) = \mathcal{O}\Big(\frac{d + \log(\frac{1}{\delta})}{n}\log^2(\frac{1}{\delta})\log n\Big).$$

*(2.) If $\frac{1}{2} < \theta \leq 1$, we suppose Assumptions 2.1, 2.4 and 2.8 hold, assume the $F_S$ satisfies Assumption 2.6 with parameter $2\mu(S)$, and suppose the $F$ satisfies Assumption 2.6 with parameter $2\mu$. For any $\delta \in (0,1)$, with probability $1 - \delta$, we have the following inequality*

$$F(\mathbf{x}_{T+1}) - F(\mathbf{x}^*) = \mathcal{O}\Big(\frac{d + \log(\frac{1}{\delta})}{n}\log^{(2\theta+1)}(\frac{1}{\delta})\log n\Big).$$

*(3.) If $\theta > 1$, we suppose Assumptions 2.1, 2.4 and 2.8 hold, , assume the $F_S$ satisfies Assumption 2.6 with parameter $2\mu(S)$, and suppose the $F$ satisfies Assumption 2.6 with parameter $2\mu$. For any $\delta \in (0,1)$, with probability $1 - \delta$, we have the following inequality*

$$F(\mathbf{x}_{T+1}) - F(\mathbf{x}^*) = \mathcal{O}\Big(\frac{d + \log(\frac{1}{\delta})}{n}\log^{(2\theta+1)}(\frac{1}{\delta})\log^{\frac{3(\theta-1)}{2}}(\frac{n}{\delta})\log n\Big).$$

*Remark* 3.8. $F(\mathbf{x}_{T+1}) - F(\mathbf{x}^*)$ measures the difference between the population risk of the last iterate and the optimal population risk. It is referred to as excess risk in learning theory (London, 2017; Feldman & Vondrak, 2019; Bassily et al., 2020). Theorem 3.7 shows that if the empirical risk and population risk satisfy the Polyak-Łojasiewicz condition, the generalization bounds of SGDM are of the order $\widetilde{\mathcal{O}}\Big(\frac{d+\log(\frac{1}{\delta})}{n}\Big)$, which improves the dependency on the sample size $n$ compared to Theorem 3.3. Due to the smoothness property in Lemma B.7: $\|\nabla F(\mathbf{x}_{T+1})\|^2 \leq (2L)(F(\mathbf{x}_{T+1}) - F(\mathbf{x}^*))$, the bounds in Theorem 3.7 also hold for $\|\nabla F(\mathbf{x}_{T+1})\|^2$. Note that in Section 3.2, the bounds provided are for the last iterate of SGDM rather than the average iterate of Section 3.1.

### 3.3 LEARNING BOUNDS WITH BERNSTEIN CONDITION

In this section, we are interested in deriving sharper generalization bounds by considering the Bernstein condition. Towards this aim, we assume that the set $\mathcal{X}$ satisfies $\mathcal{X} \subseteq B(\mathbf{x}^*, R)$.

**Theorem 3.9.** *Let $\mathbf{x}_t$ be the sequence of iterates generated by Algorithm 1. Set the stepsize as $\eta_t = \frac{1}{\mu(S)(t+t_0)}$ such that $t_0 \geq \max\{\frac{12L-4L\gamma}{\mu(S)(1-\gamma)^3}, \frac{(8C_\gamma)L}{(1-\gamma)^2\mu(S)} + 1, \frac{8C_\gamma(L\gamma+L\gamma(C_\gamma))}{(1-\gamma)\mu(S)} - 1, 1\}$, where $C_\gamma$ is a constant that depends only on $\gamma$. We choose $T \asymp n^2$.*

*(1.) If $\theta = \frac{1}{2}$, we suppose Assumptions 2.1, 2.2 and 2.8 hold, assume the $F_S$ satisfies Assumption 2.6 with parameter $2\mu(S)$, and suppose the $F$ satisfies Assumption 2.6 with parameter $2\mu$. When $n \geq \frac{cL^2(d+\log(\frac{8\log(2nR+2)}{\delta}))}{\mu^2}$, where $c$ is an absolute constant, for any $\delta \in (0,1)$, with probability $1 - \delta$, we have the following inequality*

$$F(\mathbf{x}_{T+1}) - F(\mathbf{x}^*) = \mathcal{O}\Big(\frac{\log^2(\frac{1}{\delta})}{n^2} + \frac{F(\mathbf{x}^*)\log(\frac{1}{\delta})}{n}\Big).$$

*(2.) If $\frac{1}{2} < \theta \leq 1$, we suppose Assumptions 2.1, 2.2, 2.4 and 2.8 hold, assume the $F_S$ satisfies Assumption 2.6 with parameter $2\mu(S)$, and suppose the $F$ satisfies Assumption 2.6 with parameter*

$2\mu$. When $n \geq \frac{cL^2(d+\log(\frac{8\log(2nR+2)}{\delta}))}{\mu^2}$, where $c$ is an absolute constant, for any $\delta \in (0,1)$, with probability $1 - \delta$, we have the following inequality

$$F(\mathbf{x}_{T+1}) - F(\mathbf{x}^*) = \mathcal{O}\Big(\frac{\log^{(\theta+\frac{3}{2})}(\frac{1}{\delta})\log^{\frac{1}{2}}n}{n^2} + \frac{F(\mathbf{x}^*)\log(1/\delta)}{n}\Big).$$

(3.) If $\theta > 1$, we suppose Assumptions 2.1, 2.2, 2.4 and 2.8 hold, assume the $F_S$ satisfies Assumption 2.6 with parameter $2\mu(S)$, and suppose the $F$ satisfies Assumption 2.6 with parameter $2\mu$. When $n \geq \frac{cL^2(d+\log(\frac{8\log(2nR+2)}{\delta}))}{\mu^2}$, where $c$ is an absolute constant, for any $\delta \in (0,1)$, with probability $1 - \delta$, we have the following inequality

$$F(\mathbf{x}_{T+1}) - F(\mathbf{x}^*) = \mathcal{O}\Big(\frac{\log^{\frac{3(\theta-1)}{2}}(n/\delta)\log^{(\theta+\frac{3}{2})}(\frac{1}{\delta})\log^{\frac{1}{2}}n}{n^2} + \frac{F(\mathbf{x}^*)\log(1/\delta)}{n}\Big).$$

(4.) Furthermore, assuming $F(\mathbf{x}^*) = \mathcal{O}(1/n)$, we obtain that $F(\mathbf{x}_{T+1}) - F(\mathbf{x}^*)$ is of the order $\mathcal{O}\left(\frac{\log^2(\frac{1}{\delta})}{n^2}\right)$, $\mathcal{O}\left(\frac{\log^{(\theta+\frac{3}{2})}(\frac{1}{\delta})\log^{\frac{1}{2}}n}{n^2}\right)$ and $\mathcal{O}\left(\frac{\log^{\frac{3(\theta-1)}{2}}(\frac{n}{\delta})\log^{(\theta+\frac{3}{2})}(\frac{1}{\delta})\log^{\frac{1}{2}}n}{n^2}\right)$, respectively.

*Remark* 3.10. Theorem 3.9 suggests that, under the assumptions of Theorem 3.7 and the Bernstein condition, the excess risk will be improved to $\widetilde{\mathcal{O}}\big(\frac{F(\mathbf{x}^*)}{n} + \frac{1}{n^2}\big)$. The term $F(\mathbf{x}^*)$ is tiny since it is the minimal population risk. Compared to Theorem 3.3 and Theorem 3.7, Theorem 3.9 clearly presents sharper bounds. Moreover, an obvious shortcoming of the uniform convergence approach is that it often implies learning bounds with a square-root dependency on the dimension $d$ when considering general problems (Feldman, 2016), as shown in Theorem 3.3. Another distinctive improvement of Theorem 3.9 is that we successfully remove the dimension $d$ by considering Assumption 2.2, allowing it to more easily incorporate massive neural networks that are often high-dimensional. The assumption $F(\mathbf{x}^*) = \mathcal{O}(1/n)$ we used just to show that we can get improved bounds under the low noise condition. The term $F(\mathbf{x}^*)$ should be independent of $n$. It is notable that the assumption $F(\mathbf{x}^*) = \mathcal{O}(1/n)$, or even $F(\mathbf{x}^*) = 0$, is common and can be found in (Zhang et al., 2017; Zhang & Zhou, 2019; Srebro et al., 2010; Lei et al., 2021a; Liu et al., 2018; Lei & Ying, 2020; Li & Liu, 2022). In general, the $\mathcal{O}(1/n^2)$-type generalization bounds are scarce in the learning theory community. Theorem 3.9 successfully provides $\widetilde{\mathcal{O}}(1/n^2)$ order generalization bounds with high probability for non-convex SGDM.

## 4 NUMERICAL EXPERIMENTS

We present numerical experiments to show how the generalization bound would behave versus different parameters $\theta$. Let $F_S(\mathbf{x})$ and $F_{S'}(\mathbf{x})$ be the risk built on the training dataset $S$ and the testing dataset $S'$. Thus, $F_{S'}(\mathbf{x}) = \frac{1}{|S'|}\sum_{z\in S'}f(\mathbf{x};z)$, where $|S'|$ denotes the cardinality of the set $S'$. We use $F_{S'}(\mathbf{x})$ as a good approximation of the population risk $F$. We consider six datasets available from the LIBSVM dataset: Heart, Fourclass, German, Australian, Diabetes, and Phishing (Chang & Lin, 2011). For each dataset, we take 80 percents as the training dataset and leave the remaining 20 percents as the testing dataset. According to Algorithm 1, the update of the momentum is $\mathbf{m}_t = \gamma\mathbf{m}_{t-1} + \eta_t(\nabla F_S(\mathbf{x}_t) + \nabla f(\mathbf{x}_t;z_{j_t}) - \nabla F_S(\mathbf{x}_t)) = \gamma\mathbf{m}_{t-1} + \eta_t(\nabla F_S(\mathbf{x}_t) + \mathbf{e}_t)$, where $\mathbf{e}_t = \nabla f(\mathbf{x}_t;z_{j_t}) - \nabla F_S(\mathbf{x}_t)$. In each update of the training process, for each dimension, we sample a random variable from the sub-Weibull distribution independently and identically to model the gradient noise $\mathbf{e}_t$ of Assumption 2.8. We note that if each individual entry of the random vector $\mathbf{e}_t$ follows a sub-Weibull distribution, then $\|\mathbf{e}_t\|$ is a sub-Weibull random variable. This can be proved by using Lemma 3.4 of (Bastianello et al., 2021) and part (c) of Proposition 2.1 of (Kim et al., 2021b). Since we assume that the stochastic gradient is an unbiased estimator of the exact gradient, we shift and scale the distribution in order to get a random vector with zero mean and the variance equal 1. To show the effect of the parameter $\theta$, we consider $\theta \in \{1/2, 1, 5\}$. We consider a generalized linear model $\ell(\langle\mathbf{x},x\rangle)$ for binary classification where $\ell$ is the logistic link function $\ell(s) = (1+e^{-s})^{-1}$. We first study the Huber loss, which takes the form $f(\mathbf{x},z) = \frac{1}{2}(\ell(\langle\mathbf{x},x\rangle) - y)^2$ if $|\ell(\langle\mathbf{x},x\rangle) - y| \leq \tau$ and $\tau(|\ell(\langle\mathbf{x},x\rangle) - y| - \frac{1}{2}\tau)$ otherwise. We set $\tau = 0.1$, $\gamma = 0.9$ and $\eta_t = 0.1t^{-\frac{1}{2}}$, repeat experiments 100 times, and report the average of results. The behavior of the generalization bound $\frac{1}{T}\sum_{t=1}^{T}\|\nabla F_S(\mathbf{x}_t)\|^2$ versus the number of passes is presented in Fig. 1. In our experiments, the

results are consistent with the generalization bounds of Theorem 3.3, where an increasing $\theta$ is shown to result in a worse generalization bound. When $\theta = 5$, the generalization result becomes clearly worse, which also matches the theoretical finding of the regime $\theta > 1$. Our second experiment then considers the square loss, which takes the form $f(\mathbf{x}, z) = (\ell(\langle \mathbf{x}, x \rangle) - y)^2$. In this case, the behavior of the bound $\frac{1}{T}\sum_{t=1}^{T} \|\nabla F_S(\mathbf{x}_t)\|^2$ versus the number of passes is reported in Fig. 2. Similarly, the results show that an increasing $\theta$ leads to a worse generalization bound, which is consistent with Theorem 3.3 as well.

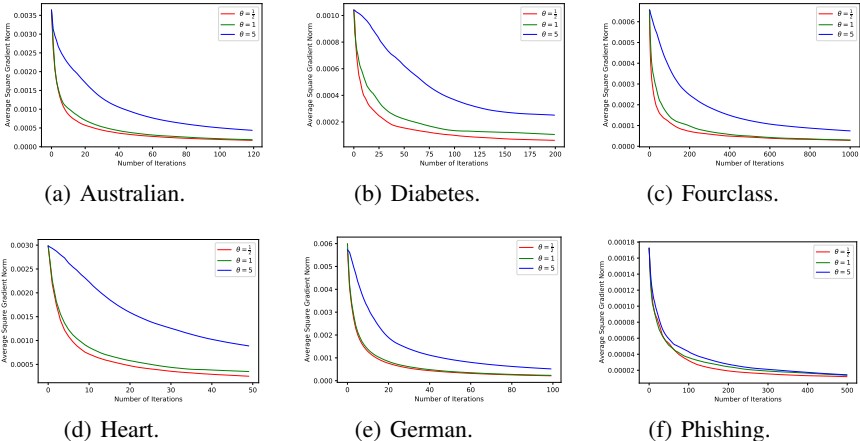

(a) Australian.  (b) Diabetes.  (c) Fourclass.

(d) Heart.  (e) German.  (f) Phishing.

Figure 1: The generalization bound $\frac{1}{T}\sum_{t=1}^{T} \|\nabla F(\mathbf{x}_t)\|^2$ versus the number of passes for different choices of $\theta \in \{1/2, 1, 5\}$ and some datasets in the setting of huber loss.

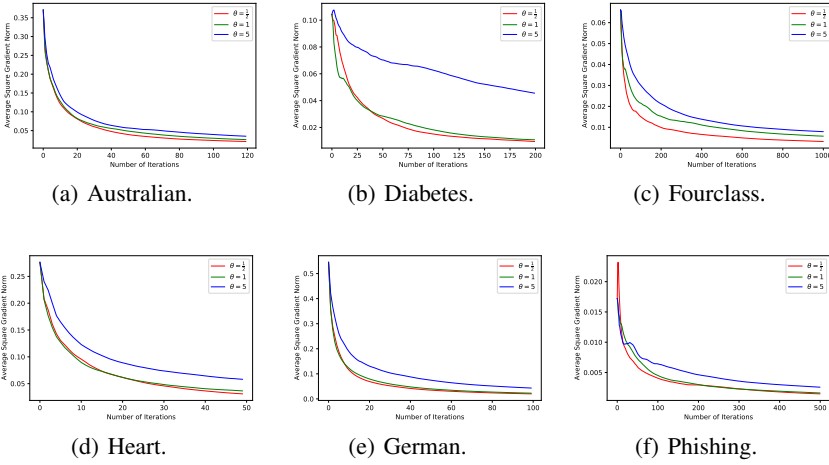

(a) Australian.  (b) Diabetes.  (c) Fourclass.

(d) Heart.  (e) German.  (f) Phishing.

Figure 2: The generalization bound $\frac{1}{T}\sum_{t=1}^{T} \|\nabla F(\mathbf{x}_t)\|^2$ versus the number of passes for different choices of $\theta \in \{1/2, 1, 5\}$ and some datasets in the setting of square loss.

## 5 CONCLUSIONS

This paper studies high probability convergence and generalization bound of stochastic gradient descent with momentum in the non-convex regime, which shows SGDM's performance in a joint view of the optimization and generalization properties. The bounds are expressed in terms of different rates and can show the impact of moving from sub-Gaussian/sub-exponential (i.e. light-tailed) variables to those with heavy tails on the rates of convergence/generalization. We believe our theoretical findings can provide deep insights into the learning guarantees of non-convex SGDM.

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

# A SUMMARY OF RESULTS

We compare the results obtained in this paper with the relevant high probability results of related work in Table 1.

Here, we provide some descriptions of Table 1. [1] is the reference (Li & Orabona, 2020), and [2] is (Cutkosky & Mehta, 2021). The second result of [1] is derived for a variant of SGDM, i.e., delayed AdaGrad with momentum whose stepsize doesn't contain the current gradient. Assumption $\theta$-*order moment* means that the gradient satisfies $\mathbb{E}_z[\|\nabla f(\mathbf{x}_t; z)\|^\theta] \leq G^\theta$ for some $G$ and $\theta \in (1, 2]$. S-S means a second-order smoothnes (Cutkosky & Mehta, 2021). There are other two convergence bounds in (Cutkosky & Mehta, 2021), derived for the last iterate of SGDM by considering the popular warm-up learning rate schedule and other tricks, see Theorem 3 and Theorem 6 in (Cutkosky & Mehta, 2021). The two bounds have the similar rate to the corresponding ones of (Cutkosky & Mehta, 2021) shown in Table 1. But their assumptions are difficult to write in a concise form, we thus omit it for brevity. *LN* means the low noise condition, i.e., $F(\mathbf{x}^*) = \mathcal{O}(1/n)$. $\theta$ corresponds to Assumption 2.8.

The comparison between our results and the results of related work has been discussed in previous Remarks. We won't repeat it here. However, one can see from Table 1 that we have provided a series of high probability generalization bounds that the related work does not involve and high probability convergence bounds with faster rates.

Table 1: Summary of Results.

| REF. | ASSUMPTION | MEASURE | LEARNING BOUND |
|------|-----------|---------|----------------|
| [1] | 2.1, $\theta = \frac{1}{2}$ | $\frac{1}{T}\sum_{t=1}^T \|\nabla F_S(\mathbf{x}_t)\|^2$ | $\mathcal{O}\left(\frac{\log(T/\delta)\log T}{\sqrt{T}}\right)$ |
| | 2.1, $\theta = \frac{1}{2}$ | $\frac{1}{T}\sum_{t=1}^T \|\nabla F_S(\mathbf{x}_t)\|^2$ | $\max\left\{\mathcal{O}\left(\frac{d\log^{\frac{3}{2}}(T/\delta)}{\sqrt{T}}\right), \mathcal{O}\left(\frac{d^2\log^2(T/\delta)}{T}\right)\right\}$ |
| [2] | $\theta$-ORDER MOMENT ($\theta \in (1,2]$), 2.1 | $\frac{1}{T}\sum_{t=1}^T \|\nabla F_S(\mathbf{x}_t)\|$ | $\mathcal{O}\left(\frac{\log(T/\delta)}{T^{\frac{\theta-1}{3\theta-2}}}\right)$ |
| | $\theta$-ORDER MOMENT ($\theta \in (1,2]$), 2.1, S-S | $\frac{1}{T}\sum_{t=1}^T \|\nabla F_S(\mathbf{x}_t)\|$ | $\mathcal{O}\left(\frac{\log(T/\delta)}{T^{\frac{2\theta-2}{5\theta-3}}}\right)$ |
| OURS | 2.1, $\theta = \frac{1}{2}$ | $\frac{1}{T}\sum_{t=1}^T \|\nabla F_S(\mathbf{x}_t)\|^2$ | $\mathcal{O}\left(\frac{\log(1/\delta)\log T}{\sqrt{T}}\right)$ |
| | 2.1, 2.4, $\theta \in (\frac{1}{2}, 1]$ | $\frac{1}{T}\sum_{t=1}^T \|\nabla F_S(\mathbf{x}_t)\|^2$ | $\mathcal{O}\left(\frac{\log^{2\theta}(1/\delta)\log T}{\sqrt{T}}\right)$ |
| | 2.1, 2.4, $\theta > 1$ | $\frac{1}{T}\sum_{t=1}^T \|\nabla F_S(\mathbf{x}_t)\|^2$ | $\mathcal{O}\left(\frac{\log^{\theta-1}(T/\delta)\log(1/\delta)+\log^{2\theta}(1/\delta)\log T}{\sqrt{T}}\right)$ |
| | 2.1, $\theta = \frac{1}{2}$ | $\frac{1}{T}\sum_{t=1}^T \|\nabla F(\mathbf{x}_t)\|^2$ | $\mathcal{O}\left(\left(\frac{d}{n}\right)^{\frac{1}{2}}\log(\frac{n}{d})\log^3(\frac{1}{\delta})\right)$ |
| | 2.1, 2.4, $\theta \in (\frac{1}{2}, 1]$ | $\frac{1}{T}\sum_{t=1}^T \|\nabla F(\mathbf{x}_t)\|^2$ | $\mathcal{O}\left(\left(\frac{d}{n}\right)^{\frac{1}{2}}\log(\frac{n}{d})\log^{(2\theta+2)}(\frac{1}{\delta})\right)$ |
| | 2.1, 2.4, $\theta > 1$ | $\frac{1}{T}\sum_{t=1}^T \|\nabla F(\mathbf{x}_t)\|^2$ | $\mathcal{O}\left(\left(\frac{d}{n}\right)^{\frac{1}{2}}\left(\log(\frac{n}{d})\log^{(2\theta+2)}(\frac{1}{\delta}) + \log^{\theta-1}(\frac{n}{d\delta})\log^2(\frac{1}{\delta})\right)\right)$ |
| | 2.1, 2.6, $\theta = \frac{1}{2}$ | $F_S(\mathbf{x}_{T+1}) - F_S(\mathbf{x}(S))$ | $\mathcal{O}\left(\frac{\log(1/\delta)}{T}\right)$ |
| | 2.1, 2.4, 2.6, $\theta \in (\frac{1}{2}, 1]$ | $F_S(\mathbf{x}_{T+1}) - F_S(\mathbf{x}(S))$ | $\mathcal{O}\left(\frac{\log^{(\theta+\frac{3}{2})}(\frac{1}{\delta})\log^{\frac{1}{2}}T}{T}\right)$ |
| | 2.1, 2.4, 2.6, $\theta > 1$ | $F_S(\mathbf{x}_{T+1}) - F_S(\mathbf{x}(S))$ | $\mathcal{O}\left(\frac{\log^{(\theta+\frac{3}{2})}(\frac{1}{\delta})\log^{\frac{3(\theta-1)}{2}}(T/\delta)\log^{\frac{1}{2}}T}{T}\right)$ |
| | 2.1, 2.6, $\theta = \frac{1}{2}$ | $F(\mathbf{x}_{T+1}) - F(\mathbf{x}^*)$ | $\mathcal{O}\left(\frac{d+\log(\frac{1}{\delta})}{n}\log^2(\frac{1}{\delta})\log n\right)$ |
| | 2.1, 2.4, 2.6, $\theta \in (\frac{1}{2}, 1]$ | $F(\mathbf{x}_{T+1}) - F(\mathbf{x}^*)$ | $\mathcal{O}\left(\frac{d+\log(\frac{1}{\delta})}{n}\log^{(2\theta+1)}(\frac{1}{\delta})\log n\right)$ |
| | 2.1, 2.4, 2.6, $\theta > 1$ | $F(\mathbf{x}_{T+1}) - F(\mathbf{x}^*)$ | $\mathcal{O}\left(\frac{d+\log(\frac{1}{\delta})}{n}\log^{(2\theta+1)}(\frac{1}{\delta})\log^{\frac{3(\theta-1)}{2}}(\frac{n}{\delta})\log n\right)$ |
| | 2.1, 2.6, 2.2, $\theta = \frac{1}{2}$ | $F(\mathbf{x}_{T+1}) - F(\mathbf{x}^*)$ | $\mathcal{O}\left(\frac{\log^2(\frac{1}{\delta})}{n^2} + \frac{F(\mathbf{x}^*)\log(\frac{1}{\delta})}{n}\right)$ |
| | 2.1, 2.4, 2.6, 2.2, $\theta \in (\frac{1}{2}, 1]$ | $F(\mathbf{x}_{T+1}) - F(\mathbf{x}^*)$ | $\mathcal{O}\left(\frac{\log^{(\theta+\frac{3}{2})}(\frac{1}{\delta})\log^{\frac{1}{2}}n}{n^2} + \frac{F(\mathbf{x}^*)\log(1/\delta)}{n}\right)$ |
| | 2.1, 2.4, 2.6, 2.2, $\theta > 1$ | $F(\mathbf{x}_{T+1}) - F(\mathbf{x}^*)$ | $\mathcal{O}\left(\frac{\log^{\frac{3(\theta-1)}{2}}(n/\delta)\log^{(\theta+\frac{3}{2})}(\frac{1}{\delta})\log^{\frac{1}{2}}n}{n^2} + \frac{F(\mathbf{x}^*)\log(1/\delta)}{n}\right)$ |
| | 2.1, 2.6, 2.2, LN, $\theta = \frac{1}{2}$ | $F(\mathbf{x}_{T+1}) - F(\mathbf{x}^*)$ | $\mathcal{O}\left(\frac{\log^2(\frac{1}{\delta})}{n^2}\right)$ |
| | 2.1, 2.4, 2.6, 2.2, LN, $\theta \in (\frac{1}{2}, 1]$ | $F(\mathbf{x}_{T+1}) - F(\mathbf{x}^*)$ | $\mathcal{O}\left(\frac{\log^{(\theta+\frac{3}{2})}(\frac{1}{\delta})\log^{\frac{1}{2}}n}{n^2}\right)$ |
| | 2.1, 2.4, 2.6, 2.2, LN, $\theta > 1$ | $F(\mathbf{x}_{T+1}) - F(\mathbf{x}^*)$ | $\mathcal{O}\left(\frac{\log^{\frac{3(\theta-1)}{2}}(n/\delta)\log^{(\theta+\frac{3}{2})}(\frac{1}{\delta})\log^{\frac{1}{2}}n}{n^2}\right)$ |

# B PRELIMINARIES

This section provides preliminaries, consisting of some properties of the Sub-Weibull distribution and some necessary auxiliary lemmas.

### B.1 SUB-WEIBULL DISTRIBUTION

Define the $L_p$ norm of random variable $X$ as $\|X\|_p = (\mathbb{E}|X|^p)^{1/p}$, for any $p \geq 1$. A sub-Weibull random variable $X$, which is denoted by $X \sim subW(\theta, K)$, can equivalently be characterized using the following properties.

**Proposition B.1.** *(Vladimirova et al., 2020; Bastianello et al., 2021) Given $\theta \geq 0$, the following properties are equivalent:*

- *$\exists K_1 > 0$ such that $P(|X| \geq t) \leq 2\exp\left(-(t/K_1)^{1/\theta}\right)$, $\forall t > 0$;*

- *$\exists K_2 > 0$ such that $\|X\|_k \leq K_2 k^\theta$, $\forall k \geq 1$;*

- *$\exists K_3 > 0$ such that $\mathbb{E}[\exp\left((\lambda|X|)^{1/\theta}\right)] \leq \exp\left((\lambda K_3)^{1/\theta}\right)$, $\forall \lambda \in (0, 1/K_3)$;*

- *$\exists K_4 > 0$ such that $\mathbb{E}\left[\exp\left((|X|/K_4)^{1/\theta}\right)\right] \leq 2$.*

*The parameters $K_1, K_2, K_3, K_4$ differ each by a constant that only depends on $\theta$.*

We introduce some concentration inequalities of sub-Weibull random variables.

**Lemma B.2.** *(Vladimirova et al., 2020; Wong et al., 2020; Madden et al., 2021) Suppose $X_1, \cdots, X_n$ are sub-Weibull($\theta$) with respective parameters $K_1, \ldots, K_n$. Then, for all $t \geq 0$,*

$$P\left(\left|\sum_{i=1}^n X_i\right| \geq t\right) \leq 2\exp\left(-\left(\frac{t}{g(\theta)\sum_{i=1}^n K_i}\right)^{1/\theta}\right),$$

*where $g(\theta) = (4e)^\theta$ for $\theta \leq 1$ and $g(\theta) = 2(2e\theta)^\theta$ for $\theta \geq 1$.*

The following two Lemmas provide concentration inequalities for the sub-Weibull martingale difference sequence.

**Lemma B.3** (Theorem 2 in (Li, 2021))**.** *(Fan & Giraudo, 2019) Let $\theta \in (0, \infty)$ be given. Assume that $(\mathbf{X}_i, i = 1, \cdots, N)$ is a sequence of $\mathbb{R}^d$-valued martingale differences with respect to filtration $\mathcal{F}_i$, i.e. $\mathbb{E}[\mathbf{X}_i|\mathcal{F}_{i-1}] = 0$, and it satisfies the following weak exponential-type tail condition: for some $\theta > 0$ and all $i = 1, ..., N$ we have for some scalar $0 < K_i$, $\mathbb{E}\left[\exp\left(\left\|\frac{\mathbf{X}_i}{K_i}\right\|^{\frac{1}{\theta}}\right)\right] \leq 2$. Assume that $K_i < \infty$ for each $i = 1, ..., N$. Then for an arbitrary $N \geq 1$ and $t > 0$,*

$$P\left(\max_{n \leq N}\left\|\sum_{i=1}^n \mathbf{X}_i\right\| \geq t\right) \leq 4\left[3 + (3\theta)^{2\theta}\frac{128\sum_{i=1}^N K_i^2}{t^2}\right]\exp\left\{-\left(\frac{t^2}{64\sum_{i=1}^N K_i^2}\right)^{\frac{1}{2\theta+1}}\right\}.$$

**Lemma B.4** (Proposition 11 in (Madden et al., 2021))**.** *[Sub-Weibull Freedman Inequality] Let $(\Omega, \mathcal{F}, (\mathcal{F}_i), P)$ be a filtered probability space. Let $(\xi_i)$ and $(K_i)$ be adapted to $(\mathcal{F}_i)$. Let $n \in \mathbb{N}$, then for all $i \in [n]$, assume $K_{i-1} \geq 0$, $\mathbb{E}[\xi_i|\mathcal{F}_{i-1}] = 0$, and*

$$\mathbb{E}\left[\exp\left((|\xi_i|/K_{i-1})^{1/\theta}\right)|\mathcal{F}_{i-1}\right] \leq 2$$

*where $\theta \geq 1/2$. If $\theta > 1/2$, assume there exists $(m_i)$ such that $K_{i-1} \leq m_i$.*

*If $\theta = 1/2$, let $a = 2$. Then for all $x, \beta \geq 0$, and $\alpha > 0$, and $\lambda \in \left[0, \frac{1}{2\alpha}\right]$,*

$$P\left(\bigcup_{k \in [n]}\left\{\sum_{i=1}^k \xi_i \geq x \text{ and } \sum_{i=1}^k aK_{i-1}^2 \leq \alpha\sum_{i=1}^k \xi_i + \beta\right\}\right) \leq \exp(-\lambda x + 2\lambda^2\beta). \quad (2)$$

*and for all $x, \beta, \lambda \geq 0$,*

$$P\left(\bigcup_{k \in [n]}\left\{\sum_{i=1}^k \xi_i \geq x \text{ and } \sum_{i=1}^k aK_{i-1}^2 \leq \beta\right\}\right) \leq \exp\left(-\lambda x + \frac{\lambda^2}{2}\beta\right).$$

If $\theta \in \left(\frac{1}{2}, 1\right]$, let $a = (4\theta)^{2\theta}e^2$ and $b = (4\theta)^\theta e$. For all $x, \beta \geq 0$, and $\alpha \geq b\max_{i\in[n]} m_i$, and $\lambda \in \left[0, \frac{1}{2\alpha}\right]$,

$$P\left(\bigcup_{k\in[n]} \left\{\sum_{i=1}^k \xi_i \geq x \text{ and } \sum_{i=1}^k aK_{i-1}^2 \leq \alpha \sum_{i=1}^k \xi_i + \beta\right\}\right) \leq \exp(-\lambda x + 2\lambda^2\beta). \quad (3)$$

and for all $x, \beta \geq 0$, and $\lambda \in \left[0, \frac{1}{b\max_{i\in[n]} m_i}\right]$,

$$P\left(\bigcup_{k\in[n]} \left\{\sum_{i=1}^k \xi_i \geq x \text{ and } \sum_{i=1}^k aK_{i-1}^2 \leq \beta\right\}\right) \leq \exp\left(-\lambda x + \frac{\lambda^2}{2}\beta\right).$$

If $\theta > 1$, let $\delta \in (0,1)$, $a = (2^{2\theta+1}+2)\Gamma(2\theta+1) + \frac{2^{3\theta}\Gamma(3\theta+1)}{3}$ and $b = 2\log^{\theta-1}(n/\delta)$. For all $x, \beta \geq 0$, and $\alpha \geq b\max_{i\in[n]} m_i$, and $\lambda \in \left[0, \frac{1}{2\alpha}\right]$,

$$P\left(\bigcup_{k\in[n]} \left\{\sum_{i=1}^k \xi_i \geq x \text{ and } \sum_{i=1}^k aK_{i-1}^2 \leq \alpha \sum_{i=1}^k \xi_i + \beta\right\}\right) \leq \exp(-\lambda x + 2\lambda^2\beta) + 2\delta. \quad (4)$$

and for all $x, \beta \geq 0$, and $\lambda \in \left[0, \frac{1}{b\max_{i\in[n]} m_i}\right]$,

$$P\left(\bigcup_{k\in[n]} \left\{\sum_{i=1}^k \xi_i \geq x \text{ and } \sum_{i=1}^k aK_{i-1}^2 \leq \beta\right\}\right) \leq \exp\left(-\lambda x + \frac{\lambda^2}{2}\beta\right) + 2\delta.$$

### B.2 AUXILIARY LEMMAS

**Lemma B.5.** *(Lei & Tang, 2021) Let $e$ be the base of the natural logarithm. There holds the following elementary inequalities.*

*(a) If $\theta \in (0,1)$, then $\sum_{k=1}^t k^{-\theta} \leq t^{1-\theta}/(1-\theta)$;*

*(b) If $\theta = 1$, then $\sum_{k=1}^t k^{-\theta} \leq \log(et)$;*

*(c) If $\theta > 1$, then $\sum_{k=1}^t k^{-\theta} \leq \frac{\theta}{\theta-1}$.*

*(d) $\sum_{k=1}^t \frac{1}{k+k_0} \leq \log(t+1)$.*

**Lemma B.6.** *(Li & Orabona, 2020) For any $T \geq 1$, it holds*

$$\sum_{t=1}^T a_t \sum_{i=1}^t b_i = \sum_{t=1}^T b_t \sum_{i=t}^T a_i \quad \text{and} \quad \sum_{t=1}^T a_t \sum_{i=0}^{t-1} b_i = \sum_{t=1}^{T-1} b_t \sum_{i=t+1}^T a_i.$$

**Lemma B.7.** *Let $\langle\cdot,\cdot\rangle$ be the inner product. Two useful properties of smoothness are shown below (Nesterov, 2014; Ward et al., 2019):*

$$g(\mathbf{x}_1) - g(\mathbf{x}_2) \leq \langle \mathbf{x}_1 - \mathbf{x}_2, \nabla g(\mathbf{x}_2)\rangle + \frac{1}{2}L\|\mathbf{x}_1 - \mathbf{x}_2\|^2,$$

$$(2L)^{-1}\|\nabla g(\mathbf{x})\|^2 \leq g(\mathbf{x}) - \inf_{\mathbf{x}} g(\mathbf{x}).$$

The following two Lemmas belong to the results of uniform convergence, which characterizes the gap between the population gradient $\nabla F$ and the empirical $\nabla F_S$. We use them to prove the generalization bounds in this paper.

**Lemma B.8** (Corollary 2 in (Lei & Tang, 2021)). *Denoted by $B_R = B(\mathbf{0}, R)$. Let $\delta \in (0,1)$ and $S = \{z_1, ..., z_n\}$ be a set of i.i.d. samples. Suppose Assumption 2.1 holds. Then with probability at least $1 - \delta$ we have*

$$\sup_{\mathbf{x}\in B_R} \|\nabla F(\mathbf{x}) - \nabla F_S(\mathbf{x})\| \leq \frac{(LR+B)}{\sqrt{n}}\left(2 + 2\sqrt{48e\sqrt{2}(\log 2 + d\log(3e))} + \sqrt{2\log(\frac{1}{\delta})}\right),$$

*where $B = \sup_{z\in\mathcal{Z}} \|\nabla f(\mathbf{0}; z)\|$ and $L$ is the smoothness argument.*

**Lemma B.9** (Lemma B.4 in (Li & Liu, 2022), (Xu & Zeevi, 2020)). *Suppose Assumptions 2.1 and 2.2 hold. Assume the population risk $F$ satisfies $F(\mathbf{x}) - F(\mathbf{x}^*) \leq \frac{1}{2\mu}\|\nabla F(\mathbf{x})\|^2$ with $\mu > 0$. For all $\mathbf{x} \in \mathcal{X} \subseteq B(\mathbf{x}^*, R)$ and any $\delta > 0$, with probability at least $1-\delta$, when $n \geq \frac{cL^2(d+\log(\frac{8\log(2nR+2)}{\delta}))}{\mu^2}$, with probability at least $1 - \delta$,*

$$\|\nabla F(\mathbf{x}) - \nabla F_S(\mathbf{x})\| \leq \|\nabla F_S(\mathbf{x})\| + \frac{\mu}{n} + \frac{2B_*\log(4/\delta)}{n} + \sqrt{\frac{8\mathbb{E}[\|\nabla f(\mathbf{x}^*; z)\|^2]\log(4/\delta)}{n}},$$

*where $c$ is an absolute constant.*

# C  PROOF OF MAIN RESULTS

## C.1  PROOF OF THEOREM 3.1

*Proof.* According to Assumption 2.1, we have

$$F_S(\mathbf{x}_{t+1}) - F_S(\mathbf{x}_t)$$

$$\leq \langle \mathbf{x}_{t+1} - \mathbf{x}_t, \nabla F_S(\mathbf{x}_t)\rangle + \frac{1}{2}L\|\mathbf{x}_{t+1} - \mathbf{x}_t\|^2 = -\langle \mathbf{m}_t, \nabla F_S(\mathbf{x}_t)\rangle + \frac{1}{2}L\|\mathbf{m}_t\|^2. \tag{5}$$

For the first term $-\langle \mathbf{m}_t, \nabla F_S(\mathbf{x}_t)\rangle$, we have

$$-\langle \mathbf{m}_t, \nabla F_S(\mathbf{x}_t)\rangle$$

$$= -\gamma\langle \mathbf{m}_{t-1}, \nabla F_S(\mathbf{x}_t)\rangle - \langle \eta_t\nabla f(\mathbf{x}_t; z_{j_t}), \nabla F_S(\mathbf{x}_t)\rangle$$

$$= -\gamma\langle \mathbf{m}_{t-1}, \nabla F_S(\mathbf{x}_{t-1})\rangle + \gamma\langle \mathbf{m}_{t-1}, \nabla F_S(\mathbf{x}_{t-1}) - \nabla F_S(\mathbf{x}_t)\rangle - \langle \eta_t\nabla f(\mathbf{x}_t; z_{j_t}), \nabla F_S(\mathbf{x}_t)\rangle$$

$$\leq -\gamma\langle \mathbf{m}_{t-1}, \nabla F_S(\mathbf{x}_{t-1})\rangle - \langle \eta_t\nabla f(\mathbf{x}_t; z_{j_t}), \nabla F_S(\mathbf{x}_t)\rangle + \gamma\|\mathbf{m}_{t-1}\|\|\nabla F_S(\mathbf{x}_{t-1}) - \nabla F_S(\mathbf{x}_t)\|$$

$$\leq -\gamma\langle \mathbf{m}_{t-1}, \nabla F_S(\mathbf{x}_{t-1})\rangle + L\gamma\|\mathbf{m}_{t-1}\|^2 - \langle \eta_t\nabla f(\mathbf{x}_t; z_{j_t}), \nabla F_S(\mathbf{x}_t)\rangle, \tag{6}$$

where the last inequality holds due to the smoothness assumption. By recurrence and using $\mathbf{m}_0 = 0$, we derive

$$-\langle \mathbf{m}_t, \nabla F_S(\mathbf{x}_t)\rangle \leq L\sum_{i=1}^{t-1}\gamma^{t-i}\|\mathbf{m}_i\|^2 - \sum_{i=1}^{t}\gamma^{t-i}\langle \eta_i\nabla f(\mathbf{x}_i; z_{j_i}), \nabla F_S(\mathbf{x}_i)\rangle. \tag{7}$$

Taking a summation from $t = 1$ to $t = T$, we get

$$F_S(\mathbf{x}_{T+1}) - F_S(\mathbf{x}_1)$$

$$\leq L\sum_{t=1}^{T}\sum_{i=1}^{t-1}\gamma^{t-i}\|\mathbf{m}_i\|^2 - \sum_{t=1}^{T}\sum_{i=1}^{t}\gamma^{t-i}\langle \eta_i\nabla f(\mathbf{x}_i; z_{j_i}), \nabla F_S(\mathbf{x}_i)\rangle + \frac{1}{2}L\sum_{t=1}^{T}\|\mathbf{m}_t\|^2. \tag{8}$$

According to Lemma B.6, we have

$$L\sum_{t=1}^{T}\sum_{i=1}^{t-1}\gamma^{t-i}\|\mathbf{m}_i\|^2 \leq L\sum_{t=1}^{T}\gamma^{-t}\|\mathbf{m}_t\|^2\sum_{i=t}^{T}\gamma^i \leq L\sum_{t=1}^{T}\gamma^{-t}\|\mathbf{m}_t\|^2\frac{\gamma^t}{1-\gamma} = \frac{L}{1-\gamma}\sum_{t=1}^{T}\|\mathbf{m}_t\|^2. \tag{9}$$

Furthermore, using Lemma B.6, we have

$$-\sum_{t=1}^{T}\sum_{i=1}^{t}\gamma^{t-i}\langle \eta_i\nabla f(\mathbf{x}_t; z_{j_i}), \nabla F_S(\mathbf{x}_i)\rangle$$

$$= -\sum_{t=1}^{T}\sum_{i=1}^{t}\gamma^{t-i}\langle \eta_i(\nabla f(\mathbf{x}_i; z_{j_i}) - \nabla F_S(\mathbf{x}_i)), \nabla F_S(\mathbf{x}_i)\rangle - \sum_{t=1}^{T}\sum_{i=1}^{t}\gamma^{t-i}\langle \eta_i(\nabla F_S(\mathbf{x}_i)), \nabla F_S(\mathbf{x}_i)\rangle$$

$$= -\sum_{t=1}^{T}\gamma^{-t}\langle \eta_t(\nabla f(\mathbf{x}_t; z_{j_t}) - \nabla F_S(\mathbf{x}_t)), \nabla F_S(\mathbf{x}_t)\rangle\sum_{i=t}^{T}\gamma^i - \sum_{t=1}^{T}\gamma^{-t}\langle \eta_t(\nabla F_S(\mathbf{x}_t)), \nabla F_S(\mathbf{x}_t)\rangle\sum_{t=1}^{T}\gamma^i$$

$$\leq -\sum_{t=1}^{T}\gamma^{-t}\langle \eta_t(\nabla f(\mathbf{x}_t; z_{j_t}) - \nabla F_S(\mathbf{x}_t)), \nabla F_S(\mathbf{x}_t)\rangle\sum_{i=t}^{T}\gamma^i - \sum_{t=1}^{T}\eta_t\|\nabla F_S(\mathbf{x}_t)\|^2$$

$$= -\sum_{t=1}^{T}\frac{1-\gamma^{T-t+1}}{1-\gamma}\langle \eta_t(\nabla f(\mathbf{x}_t; z_{j_t}) - \nabla F_S(\mathbf{x}_t)), \nabla F_S(\mathbf{x}_t)\rangle - \sum_{t=1}^{T}\eta_t\|\nabla F_S(\mathbf{x}_t)\|^2. \tag{10}$$

Plugging (9) and (10) into (8), we obtain

$$
\sum_{t=1}^{T} \eta_t \|\nabla F_S(\mathbf{x}_t))\|^2 \leq F_S(\mathbf{x}_1) - F_S(\mathbf{x}_S) + \frac{L}{1-\gamma} \sum_{t=1}^{T} \|\mathbf{m}_t\|^2
$$

$$
- \sum_{t=1}^{T} \frac{1-\gamma^{T-t+1}}{1-\gamma} \langle \eta_t(\nabla f(\mathbf{x}_t; z_{j_t}) - \nabla F_S(\mathbf{x}_t)), \nabla F_S(\mathbf{x}_t)\rangle + \frac{1}{2}L \sum_{t=1}^{T} \|\mathbf{m}_t\|^2. \qquad (11)
$$

It is clear that

$$
\mathbb{E}_{j_t} \left[ -\frac{1-\gamma^{T-t+1}}{1-\gamma} \langle \eta_t(\nabla f(\mathbf{x}_t; z_{j_t}) - \nabla F_S(\mathbf{x}_t)), \nabla F_S(\mathbf{x}_t)\rangle \right] = 0,
$$

implying that it is a martingale difference sequence (MDS). We thus use Lemma B.4 to bound it. Specifically, we set $\xi_t = -\frac{1-\gamma^{T-t+1}}{1-\gamma} \langle \eta_t(\nabla f(\mathbf{x}_t; z_{j_t}) - \nabla F_S(\mathbf{x}_t)), \nabla F_S(\mathbf{x}_t)\rangle$, $K_{t-1} = \frac{1-\gamma^{T-t+1}}{1-\gamma} \eta_t K \|\nabla F_S(\mathbf{x}_t)\|$, $\beta = 0$, $\lambda = \frac{1}{2\alpha}$, and $x = 2\alpha \log(1/\delta)$.

If $\theta = \frac{1}{2}$, for all $\alpha > 0$, we have the following inequality with probability $1 - \delta$

$$
- \sum_{t=1}^{T} \frac{1-\gamma^{T-t+1}}{1-\gamma} \langle \eta_t(\nabla f(\mathbf{x}_t; z_{j_t}) - \nabla F_S(\mathbf{x}_t)), \nabla F_S(\mathbf{x}_t)\rangle
$$

$$
\leq 2\alpha \log(1/\delta) + \frac{aK^2}{\alpha} \sum_{t=1}^{T} \eta_t^2 \left(\frac{1-\gamma^{T-t+1}}{1-\gamma}\right)^2 \|\nabla F_S(\mathbf{x}_t)\|^2
$$

$$
\leq 2\alpha \log(1/\delta) + \frac{aK^2}{\alpha} \left(\frac{1-\gamma^T}{1-\gamma}\right)^2 \sum_{t=1}^{T} \eta_t^2 \|\nabla F_S(\mathbf{x}_t)\|^2.
$$

If $\theta \in (\frac{1}{2}, 1]$, according to Assumption 2.4, we set $m_t = \frac{1-\gamma^T}{1-\gamma} KG$. Then for all $\alpha \geq b\frac{1-\gamma^T}{1-\gamma} KG$, we have the following inequality with probability $1 - \delta$

$$
- \sum_{t=1}^{T} \frac{1-\gamma^{T-t+1}}{1-\gamma} \langle \eta_t(\nabla f(\mathbf{x}_t; z_{j_t}) - \nabla F_S(\mathbf{x}_t)), \nabla F_S(\mathbf{x}_t)\rangle
$$

$$
\leq 2\alpha \log(1/\delta) + \frac{aK^2}{\alpha} \left(\frac{1-\gamma^T}{1-\gamma}\right)^2 \sum_{t=1}^{T} \eta_t^2 \|\nabla F_S(\mathbf{x}_t)\|^2.
$$

If $\theta > 1$, according to Assumption 2.4, we set $m_t = \frac{1-\gamma^T}{1-\gamma} KG$ and $\delta = \delta$. Then, for all $\alpha \geq b\frac{1-\gamma^T}{1-\gamma} KG$, we have the following inequality with probability $1 - 3\delta$

$$
- \sum_{t=1}^{T} \frac{1-\gamma^{T-t+1}}{1-\gamma} \langle \eta_t(\nabla f(\mathbf{x}_t; z_{j_t}) - \nabla F_S(\mathbf{x}_t)), \nabla F_S(\mathbf{x}_t)\rangle
$$

$$
\leq 2\alpha \log(1/\delta) + \frac{aK^2}{\alpha} \left(\frac{1-\gamma^T}{1-\gamma}\right)^2 \sum_{t=1}^{T} \eta_t^2 \|\nabla F_S(\mathbf{x}_t)\|^2.
$$

Then, we consider the term $\sum_{t=1}^{T} \|\mathbf{m}_t\|^2$.

$$
\begin{aligned}
\sum_{t=1}^{T} \|\mathbf{m}_t\|^2 &= \sum_{t=1}^{T} \left\| \gamma \mathbf{m}_{t-1} + (1-\gamma) \frac{\eta_t \nabla f(\mathbf{x}_t; z_{j_t})}{1-\gamma} \right\|^2 \\
&\leq \sum_{t=1}^{T} \left( \gamma \|\mathbf{m}_{t-1}\|^2 + (1-\gamma) \left\| \frac{\eta_t \nabla f(\mathbf{x}_t; z_{j_t})}{1-\gamma} \right\|^2 \right) \\
&= \sum_{t=1}^{T-1} \gamma \|\mathbf{m}_t\|^2 + \sum_{t=1}^{T} (1-\gamma) \left\| \frac{\eta_t \nabla f(\mathbf{x}_t; z_{j_t})}{1-\gamma} \right\|^2 ) \\
&\leq \sum_{t=1}^{T} \gamma \|\mathbf{m}_t\|^2 + \sum_{t=1}^{T} (1-\gamma) \left\| \frac{\eta_t \nabla f(\mathbf{x}_t; z_{j_t})}{1-\gamma} \right\|^2 ),
\end{aligned}
$$

where the first inequality holds due to the Jensen's inequality and the second equality follows from $\|\mathbf{m}_0\| = 0$. Thus, we have

$$
\sum_{t=1}^{T} \|\mathbf{m}_t\|^2 \leq \sum_{t=1}^{T} \frac{1}{(1-\gamma)^2} \|\eta_t \nabla f(\mathbf{x}_t; z_{j_t})\|^2. \tag{12}
$$

Then we have

$$
\sum_{t=1}^{T} \|\mathbf{m}_t\|^2 \leq \frac{2}{(1-\gamma)^2} \sum_{t=1}^{T} \eta_t^2 \|\nabla f(\mathbf{x}_t; z_{j_t}) - \nabla F_S(\mathbf{x}_t)\|^2 + \frac{2}{(1-\gamma)^2} \sum_{t=1}^{T} \eta_t^2 \|\nabla F_S(\mathbf{x}_t)\|^2.
$$

Since $\|\nabla f(\mathbf{x}_t; z_{j_t}) - \nabla F_S(\mathbf{x}_t)\|$ is a sub-Weibull random variable, we get

$$
\mathbb{E}\left[ \exp \left( \frac{\eta_t^2 \|\nabla f(\mathbf{x}_t; z_{j_t}) - \nabla F_S(\mathbf{x}_t)\|^2}{\eta_t^2 K^2} \right)^{\frac{1}{2\theta}} \right] \leq 2,
$$

which means that $\eta_t^2 \|\nabla f(\mathbf{x}_t; z_{j_t}) - \nabla F_S(\mathbf{x}_t)\|^2 \sim subW(2\theta, \eta_t^2 K^2)$. According to Lemma B.2, we get the following inequality with probability $1 - \delta$

$$
\sum_{t=1}^{T} \frac{2}{(1-\gamma)^2} \eta_t^2 \|\nabla f(\mathbf{x}_t; z_{j_t}) - \nabla F_S(\mathbf{x}_t)\|^2 \leq \frac{2}{(1-\gamma)^2} K^2 g(2\theta) \log^{2\theta}(2/\delta) \sum_{t=1}^{T} \eta_t^2.
$$

Then, we plug the bound of $-\sum_{t=1}^{T} \frac{1-\gamma^{T-t+1}}{1-\gamma} \langle \eta_t(\nabla f(\mathbf{x}_t; z_{j_t}) - \nabla F_S(\mathbf{x}_t)), \nabla F_S(\mathbf{x}_t) \rangle$ and the bound of $\sum_{t=1}^{T} \|\mathbf{m}_t\|^2$ into (11), we obtain

$$
\begin{aligned}
\sum_{t=1}^{T} \eta_t \|\nabla F_S(\mathbf{x}_t)\|^2 &\leq F_S(\mathbf{x}_1) - F_S(\mathbf{x}(S)) + \left( \frac{L}{1-\gamma} + \frac{1}{2}L \right) \frac{2}{(1-\gamma)^2} \sum_{t=1}^{T} \eta_t^2 \|\nabla F_S(\mathbf{x}_t)\|^2 \\
&+ 2\alpha \log(1/\delta) + \frac{aK^2}{\alpha} \left( \frac{1-\gamma^T}{1-\gamma} \right)^2 \sum_{t=1}^{T} \eta_t^2 \|\nabla F_S(\mathbf{x}_t)\|^2 \\
&+ \left( \frac{L}{1-\gamma} + \frac{1}{2}L \right) \frac{2}{(1-\gamma)^2} K^2 g(2\theta) \log^{2\theta}(2/\delta) \sum_{t=1}^{T} \eta_t^2,
\end{aligned}
$$

implying that

$$
\begin{aligned}
&\sum_{t=1}^{T} \eta_t \left( 1 - \left( \frac{L}{1-\gamma} + \frac{1}{2}L \right) \frac{2}{(1-\gamma)^2} \eta_t - \frac{aK^2}{\alpha} \left( \frac{1-\gamma^T}{1-\gamma} \right)^2 \eta_t \right) \|\nabla F_S(\mathbf{x}_t)\|^2 \\
&\leq F_S(\mathbf{x}_1) - F_S(\mathbf{x}_S) + 2\alpha \log(1/\delta) + \left( \frac{L}{1-\gamma} + \frac{1}{2}L \right) \frac{2}{(1-\gamma)^2} K^2 g(2\theta) \log^{2\theta}(2/\delta) \sum_{t=1}^{T} \eta_t^2.
\end{aligned}
$$

When $c = \eta_1 \leq \frac{1}{8} \frac{(1-\gamma)^2}{\frac{L}{1-\gamma} + \frac{1}{2}L} = \frac{1}{4} \frac{(1-\gamma)^3}{3L - L\gamma}$, then

$$\left(\frac{L}{1-\gamma} + \frac{1}{2}L\right) \frac{2}{(1-\gamma)^2} \eta_t \leq \frac{1}{4}, \forall t. \tag{13}$$

When $\frac{aK^2}{\alpha}(\frac{1-\gamma^T}{1-\gamma})^2 \eta_t \leq \frac{1}{4}$, then

$$\alpha \geq 4\left(\frac{1-\gamma^T}{1-\gamma}\right)^2 \eta_1 aK^2.$$

Thus, if $\alpha \geq 4(\frac{1-\gamma^T}{1-\gamma})^2 \eta_1 aK^2 = 4(\frac{1-\gamma^T}{1-\gamma})^2 caK^2$ and $\eta_1 \leq \frac{1}{8} \frac{(1-\gamma))^2}{\frac{L}{1-\gamma} + \frac{1}{2}L}$, we derive that

$$\sum_{t=1}^{T} \eta_t \|\nabla F_S(\mathbf{x}_t)\|^2$$

$$\leq 2(F_S(\mathbf{x}_1) - F_S(\mathbf{x}(S))) + 4\alpha \log(1/\delta) + 2\left(\frac{L}{1-\gamma} + \frac{1}{2}L\right) \frac{2}{(1-\gamma)^2} K^2 g(2\theta) \log^{2\theta}(2/\delta) \sum_{t=1}^{T} \eta_t^2.$$

Putting the previous bounds together. Hence, if $\theta = \frac{1}{2}$, taking $\alpha = 4(\frac{1-\gamma^T}{1-\gamma})^2 \eta_1 aK^2 = 8(\frac{1-\gamma^T}{1-\gamma})^2 \eta_1 K^2$, with probability $1 - 2\delta$, we have

$$\sum_{t=1}^{T} \eta_t \|\nabla F_S(\mathbf{x}_t)\|^2 \leq 2(F_S(\mathbf{x}_1) - F_S(\mathbf{x}(S))) + 32\left(\frac{1-\gamma^T}{1-\gamma}\right)^2 \eta_1 K^2 \log(1/\delta)$$

$$+ \left(\frac{L}{1-\gamma} + \frac{1}{2}L\right) \frac{4}{(1-\gamma)^2} K^2 g(1) \log(2/\delta) \sum_{t=1}^{T} \eta_t^2.$$

If $\frac{1}{2} < \theta \leq 1$, taking $\alpha = \max\left\{b\frac{1-\gamma^T}{1-\gamma}KG, 4(\frac{1-\gamma^T}{1-\gamma})^2 \eta_1 aK^2\right\}$
$= \max\left\{(4\theta)^\theta e\frac{1-\gamma^T}{1-\gamma}KG, 4(\frac{1-\gamma^T}{1-\gamma})^2 \eta_1 (4\theta)^{2\theta} e^2 K^2\right\}$, with probability $1 - 2\delta$, we have

$$\sum_{t=1}^{T} \eta_t \|\nabla F_S(\mathbf{x}_t)\|^2 \leq 2(F_S(\mathbf{x}_1) - F_S(\mathbf{x}(S)))$$

$$+ 4\max\left\{(4\theta)^\theta e\frac{1-\gamma^T}{1-\gamma}KG, 4(\frac{1-\gamma^T}{1-\gamma})^2 \eta_1 (4\theta)^{2\theta} e^2 K^2\right\} \log(\frac{1}{\delta})$$

$$+ \left(\frac{L}{1-\gamma} + \frac{1}{2}L\right) \frac{4}{(1-\gamma)^2} K^2 g(2\theta) \log^{2\theta}(2/\delta) \sum_{t=1}^{T} \eta_t^2.$$

If $\theta > 1$, taking $\alpha = \max\left\{b\frac{1-\gamma^T}{1-\gamma}KG, 4(\frac{1-\gamma^T}{1-\gamma})^2 \eta_1 aK^2\right\}$, that is

$$\alpha = \max\left\{2\log^{\theta-1}(T/\delta)\frac{1-\gamma^T}{1-\gamma}KG, 4(\frac{1-\gamma^T}{1-\gamma})^2 \eta_1((2^{2\theta+1} + 2)\Gamma(2\theta + 1) + \frac{2^{3\theta}\Gamma(3\theta + 1)}{3})K^2\right\}.$$

Thus, with probability $1 - 4\delta$, we have

$$\sum_{t=1}^{T} \eta_t \|\nabla F_S(\mathbf{x}_t)\|^2 \leq 2(F_S(\mathbf{x}_1) - F_S(\mathbf{x}(S)))$$

$$+ \left(\frac{L}{1-\gamma} + \frac{1}{2}L\right) \frac{4}{(1-\gamma)^2} K^2 g(2\theta) \log^{2\theta}(2/\delta) \sum_{t=1}^{T} \eta_t^2$$

$$+ 4\log(1/\delta) \max\left\{2\log^{\theta-1}(T/\delta)\frac{1-\gamma^T}{1-\gamma}KG,\right.$$

$$\left. 4(\frac{1-\gamma^T}{1-\gamma})^2 \eta_1((2^{2\theta+1} + 2)\Gamma(2\theta + 1) + \frac{2^{3\theta}\Gamma(3\theta + 1)}{3})K^2\right\}.$$

Note that the dependence on confidence parameter $1/\delta$ in above bounds is logarithmic. One can replace $\delta$ to $\delta/2$ or $\delta/4$. Through this simple transformation, we have the following results: (1.) if $\theta = 1$, under Assumptions 2.1 and 2.8, with probability $1 - \delta$, we have

$$\frac{1}{T}\sum_{t=1}^{T}\|\nabla F_S(\mathbf{x}_t)\|^2 \leq \frac{1}{c\sqrt{T}}\sum_{t=1}^{T}\eta_t\|\nabla F_S(\mathbf{x}_t)\|^2 = \mathcal{O}\left(\frac{1}{\sqrt{T}}\log(1/\delta)\sum_{t=1}^{T}\eta_t^2\right)$$

$$= \mathcal{O}\left(\frac{1}{\sqrt{T}}\log(1/\delta)\log T\right); \tag{14}$$

(2.) if $\frac{1}{2} < \theta \leq 1$, under Assumptions 2.1, 2.4, and 2.8, with probability $1 - \delta$, we have

$$\frac{1}{T}\sum_{t=1}^{T}\|\nabla F_S(\mathbf{x}_t)\|^2 \leq \frac{1}{c\sqrt{T}}\sum_{t=1}^{T}\eta_t\|\nabla F_S(\mathbf{x}_t)\|^2 = \mathcal{O}\left(\frac{1}{\sqrt{T}}\log^{2\theta}(1/\delta)\sum_{t=1}^{T}\eta_t^2\right)$$

$$= \mathcal{O}\left(\frac{1}{\sqrt{T}}\log^{2\theta}(1/\delta)\log T\right); \tag{15}$$

(3.) if $\theta > 1$, under Assumptions 2.1, 2.4, and 2.8, with probability $1 - \delta$, we have

$$\frac{1}{T}\sum_{t=1}^{T}\|\nabla F_S(\mathbf{x}_t)\|^2 \leq \frac{1}{c\sqrt{T}}\sum_{t=1}^{T}\eta_t\|\nabla F_S(\mathbf{x}_t)\|^2$$

$$= \mathcal{O}\left(\frac{\log^{\theta-1}(T/\delta)\log(1/\delta) + \log^{2\theta}(1/\delta)\sum_{t=1}^{T}\eta_t^2}{\sqrt{T}}\right)$$

$$= \mathcal{O}\left(\frac{\log^{\theta-1}(T/\delta)\log(1/\delta) + \log^{2\theta}(1/\delta)\log T}{\sqrt{T}}\right), \tag{16}$$

where the bound of $\sum_{t=1}^{T}\eta_t^2$ follows from Lemma B.5. The proof is complete. $\qquad\square$

## C.2 PROOF OF THEOREM 3.3

*Proof.* The proof is divided into three parts.

**(1.)** In the first part, we prove the bound of $\|\mathbf{x}_t\|$. $\|\mathbf{x}_t\|$ characterizes the bound of $B(\mathbf{0}, R)$, i.e., at iterate $t$, $R = R_t = \|\mathbf{x}_t\|$, because $\mathbf{x}_t$ traverses over a ball with an increasing radius as $t$ increases. Therefore one should apply Lemma B.8 with an increasing $R$.

Since $\mathbf{x}_{t+1} = \mathbf{x}_t - \mathbf{m}_t$, by a summation and using $\mathbf{m}_1 = 0$, we get $\mathbf{x}_{t+1} = -\sum_{i=1}^{t}\mathbf{m}_i$. Since $\mathbf{m}_i = \gamma\mathbf{m}_{i-1} + \eta_i\nabla f(\mathbf{x}_i; z_{j_i})$, by recurrence, we have

$$\mathbf{m}_i = \sum_{k=1}^{i}\gamma^{i-k}\eta_k\nabla f(\mathbf{x}_k; z_{j_k}).$$

According to Lemma B.6, this gives that

$$\mathbf{x}_{t+1} = -\sum_{i=1}^{t}\sum_{k=1}^{i}\gamma^{i-k}\eta_k\nabla f(\mathbf{x}_k; z_{j_k}) = -\sum_{i=1}^{t}\frac{1-\gamma^{t-i+1}}{1-\gamma}\eta_i\nabla f(\mathbf{x}_i; z_{j_i}). \tag{17}$$

Thus, we have

$$\|\mathbf{x}_{t+1}\| = \frac{1}{1-\gamma}\left\|\sum_{i=1}^{t}(1-\gamma^{t-i+1})\eta_i\nabla f(\mathbf{x}_i; z_{j_i})\right\|$$

$$\leq \frac{1}{1-\gamma}\left\|\sum_{i=1}^{t}(1-\gamma^{t-i+1})\eta_i(\nabla f(\mathbf{x}_i; z_{j_i}) - \nabla F_S(\mathbf{x}_i))\right\| + \frac{1}{1-\gamma}\left\|\sum_{i=1}^{t}(1-\gamma^{t-i+1})\eta_i\nabla F_S(\mathbf{x}_i)\right\|$$

$$\leq \frac{1}{1-\gamma}\left\|\sum_{i=1}^{t}(1-\gamma^{t-i+1})\eta_i(\nabla f(\mathbf{x}_i; z_{j_i}) - \nabla F_S(\mathbf{x}_i))\right\| + \frac{1}{1-\gamma}\left\|\sum_{i=1}^{t}(1-\gamma^t)\eta_i\nabla F_S(\mathbf{x}_i)\right\|. \tag{18}$$

Let's consider the first term $\left\| \sum_{i=1}^{t}(1 - \gamma^{t-i+1})\eta_i(\nabla f(\mathbf{x}_i; z_{j_i}) - \nabla F_S(\mathbf{x}_i)) \right\|$. It is clear that $\mathbb{E}_{j_i}[(1 - \gamma^{t-i+1})\eta_i(\nabla f(\mathbf{x}_i; z_{j_i}) - \nabla F_S(\mathbf{x}_i))] = 0$, which means that it is a MDS. Moreover, since $\|\nabla f(\mathbf{x}_i; z_{j_i}) - \nabla F_S(\mathbf{x}_i)\| \sim subW(\theta, K)$, we have

$$\mathbb{E}\left[\exp\left(\frac{\|\eta_i(1 - \gamma^{t-i+1})(\nabla f(\mathbf{x}_i; z_{j_i}) - \nabla F_S(\mathbf{x}_i))\|}{\eta_i(1 - \gamma^t)K}\right)^{\frac{1}{\theta}}\right] \leq 2.$$

Then, we can apply Lemma B.3 to derive the following inequality

$$P\left(\max_{1 \leq t \leq T}\left\|\sum_{i=1}^{t}(1 - \gamma^{t-i+1})\eta_i(\nabla f(\mathbf{x}_i; z_{j_i}) - \nabla F_S(\mathbf{x}_i))\right\| \geq x\right)$$

$$\leq 4\left[3 + (3\theta)^{2\theta}\frac{128K^2(1 - \gamma^T)\sum_{i=1}^{T}\eta_i^2}{x^2}\right]\exp\left\{-\left(\frac{x^2}{64K^2(1 - \gamma^T)\sum_{i=1}^{T}\eta_i^2}\right)^{\frac{1}{2\theta+1}}\right\}.$$

Setting the term $4\exp\left\{-\left(\frac{x^2}{64K^2(1-\gamma^T)\sum_{i=1}^{T}\eta_i^2}\right)^{\frac{1}{2\theta+1}}\right\}$ equal to $\delta$, we get $x = 8\log^{(\theta+\frac{1}{2})}(\frac{4}{\delta})K(1 - \gamma^T)^{\frac{1}{2}}(\sum_{i=1}^{T}\eta_i^2)^{\frac{1}{2}}$. Thus, with probability $1 - 3\delta - \frac{8(3\theta)^{2\theta}}{\log^{2\theta+1}\frac{4}{\delta}}\delta$, we have

$$\max_{1 \leq t \leq T}\left\|\sum_{i=1}^{t}\eta_i(\nabla f(\mathbf{w}_i; z_{j_i}) - \nabla F_S(\mathbf{w}_i))\right\| \leq 8\log^{(\theta+\frac{1}{2})}(\frac{4}{\delta})K(1 - \gamma^T)^{\frac{1}{2}}\left(\sum_{i=1}^{T}\eta_i^2\right)^{\frac{1}{2}}. \quad (19)$$

Since $\theta \geq 1/2$ and $\delta \in (0, 1)$, we have $\log^{2\theta+1}\frac{4}{\delta} > 1$. Thus, (19) means that with probability $1 - 3\delta - 8(3\theta)^{2\theta}\delta$, we have

$$\max_{1 \leq t \leq T}\left\|\sum_{i=1}^{t}\eta_i(\nabla f(\mathbf{w}_i; z_{j_i}) - \nabla F_S(\mathbf{w}_i))\right\| \leq 8\log^{(\theta+\frac{1}{2})}(\frac{4}{\delta})K(1 - \gamma^T)^{\frac{1}{2}}\left(\sum_{i=1}^{T}\eta_i^2\right)^{\frac{1}{2}}.$$

Now, with probability $1 - \delta$, we can derive

$$\max_{1 \leq t \leq T}\left\|\sum_{i=1}^{t}\eta_i(\nabla f(\mathbf{w}_i; z_{j_i}) - \nabla F_S(\mathbf{w}_i))\right\|$$

$$\leq 8\log^{(\theta+\frac{1}{2})}\left(\frac{4(3 + 8(3\theta)^{2\theta})}{\delta}\right)K(1 - \gamma^T)^{\frac{1}{2}}\left(\sum_{i=1}^{T}\eta_i^2\right)^{\frac{1}{2}}. \quad (20)$$

For the second term $\left\|\sum_{i=1}^{t}\eta_i\nabla F_S(\mathbf{x}_i)\right\|$, we have

$$\left\|\sum_{i=1}^{t}\eta_i\nabla F_S(\mathbf{x}_i)\right\|^2 \leq \left(\sum_{i=1}^{t}\eta_i\|\nabla F_S(\mathbf{x}_i)\|\right)^2 \leq \left(\sum_{i=1}^{t}\eta_i\right)\left(\sum_{i=1}^{t}\eta_i\|\nabla F_S(\mathbf{x}_i)\|^2\right). \quad (21)$$

where the second inequality follows form the Schwarz's inequality. For the sake of the presentation, we introduce a notation $\Delta(\theta, T, \delta) = \log^{\theta-1}(T/\delta)\log(1/\delta)\mathbb{I}_{\theta>1}$, where $\mathbb{I}_{\theta>1}$ is an indication function. Thus with probability $1 - \delta$ we have the following inequality uniformly for all $t = 1, ..., T$

$$\left\|\sum_{i=1}^{t}\eta_i\nabla F_S(\mathbf{x}_i)\right\|^2 \leq \left(\sum_{i=1}^{t}\eta_i\right)\left(\sum_{i=1}^{t}\eta_i\|\nabla F_S(\mathbf{x}_i)\|^2\right)$$

$$= \left(\sum_{i=1}^{t}\eta_i\right)\mathcal{O}\left(\Delta(\theta, T, \delta) + \log^{2\theta}(1/\delta)\sum_{i=1}^{t}\eta_i^2\right), \quad (22)$$

where the last equation follows from the results of (14), (15), and (16).

Plugging (20) and (22) into (18), we have the following inequality uniformly for all $t = 1, ..., T$ with probability at least $1 - 2\delta$

$$\|\mathbf{x}_{t+1}\| = \mathcal{O}\left(\log^{(\theta+\frac{1}{2})}(\frac{1}{\delta})(1-\gamma^T)^{\frac{1}{2}}(\sum_{i=1}^{T}\eta_i^2)^{\frac{1}{2}}\right) + \left(\left(\sum_{i=1}^{t}\eta_i\right)\mathcal{O}\left(\Delta(\theta,T,\delta)+\log^{2\theta}(1/\delta)\sum_{i=1}^{t}\eta_i^2\right)\right)^{\frac{1}{2}}$$
(23)

$$= \mathcal{O}\left(\log^{(\theta+\frac{1}{2})}(\frac{1}{\delta})(1-\gamma^T)^{\frac{1}{2}}\log^{\frac{1}{2}}T\right) + \left(t^{\frac{1}{2}}\mathcal{O}\left(\Delta(\theta,T,\delta)+\log^{2\theta}(1/\delta)\log t\right)\right)^{\frac{1}{2}}$$

$$\leq \mathcal{O}\left(t^{\frac{1}{4}}\left(\Delta^{\frac{1}{2}}(\theta,T,\delta)+\log^{(\theta+\frac{1}{2})}(\frac{1}{\delta})\log^{\frac{1}{2}}T\right)\right),$$
(24)

where the second equation follows from Lemma B.5.

**(2.)** In the second part, we prove the bound of $\max_{1\leq t\leq T}\|\nabla F(\mathbf{x}_t) - \nabla F_S(\mathbf{x}_t)\|$. According to Lemma B.8, with probability $1 - \delta$ we have

$$\max_{1\leq t\leq T}\|\nabla F(\mathbf{x}_t) - \nabla F_S(\mathbf{x}_t)\|$$

$$\leq \frac{(LR_T + B)}{\sqrt{n}}\left(2 + 2\sqrt{48e\sqrt{2}(\log 2 + d\log(3e))} + \sqrt{2\log(\frac{1}{\delta})}\right)$$

$$\leq \frac{(L\|\mathbf{x}_T\| + B)}{\sqrt{n}}\left(2 + 2\sqrt{48e\sqrt{2}(\log 2 + d\log(3e))} + \sqrt{2\log(\frac{1}{\delta})}\right).$$
(25)

Plugging (24) into (25), with probability $1 - 3\delta$ we have the following inequality uniformly for all $t = 1, ...T$

$$\max_{1\leq t\leq T}\|\nabla F(\mathbf{x}_t) - \nabla F_S(\mathbf{x}_t)\| \leq$$

$$\frac{L\mathcal{O}\left(T^{\frac{1}{4}}\left(\Delta^{\frac{1}{2}}(\theta,T,\delta)+\log^{(\theta+\frac{1}{2})}(\frac{1}{\delta})\log^{\frac{1}{2}}T\right)\right)+B}{\sqrt{n}}\left(2+2\sqrt{48e\sqrt{2}(\log 2 + d\log(3e))}+\sqrt{2\log(\frac{1}{\delta})}\right),$$

which means that we have the following inequality uniformly for all $t = 1, ...T$ with probability $1 - \delta$

$$\max_{1\leq t\leq T}\|\nabla F(\mathbf{x}_t) - \nabla F_S(\mathbf{x}_t)\|^2$$

$$= \mathcal{O}\left(\frac{T^{\frac{1}{2}}\left(\Delta(\theta,T,\delta)+\log^{(2\theta+1)}(\frac{1}{\delta})\log T\right)}{n}\times\left(d+\log(\frac{1}{\delta})\right)\right).$$
(26)

**(3.)** In the third part, we prove the bound of $\frac{1}{T}\sum_{t=1}^{T}\|\nabla F(\mathbf{x}_t)\|^2$. Firstly, we can derive the following inequality with probability $1 - 2\delta$

$$\sum_{t=1}^{T}\eta_t\|\nabla F(\mathbf{x}_t)\|^2$$

$$\leq 2\sum_{t=1}^{T}\eta_t\|\nabla F(\mathbf{x}_t) - \nabla F_S(\mathbf{x}_t)\|^2 + 2\sum_{t=1}^{T}\eta_t\|\nabla F_S(\mathbf{x}_t)\|^2$$

$$\leq 2\sum_{t=1}^{T}\eta_t\max_{1\leq t\leq T}\|\nabla F(\mathbf{x}_t) - \nabla F_S(\mathbf{x}_t)\|^2 + 2\sum_{t=1}^{T}\eta_t\|\nabla F_S(\mathbf{x}_t)\|^2$$

$$\leq 2\sum_{t=1}^{T}\eta_t\mathcal{O}\left(\frac{T^{\frac{1}{2}}\left(\Delta(\theta,T,\delta)+\log^{(2\theta+1)}(\frac{1}{\delta})\log T\right)}{n}\left(d+\log(\frac{1}{\delta})\right)\right)$$

$$+ \mathcal{O}\left(\Delta(\theta,T,\delta)+\log^{2\theta}(1/\delta)\log T\right),$$

where the last inequality follows from (26) and the results of (14), (15), and (16).

Therefore, we have

$$\frac{1}{T}\sum_{t=1}^{T}\|\nabla F(\mathbf{x}_t)\|^2 \le \frac{1}{c\sqrt{T}}\sum_{t=1}^{T}\eta_t\|\nabla F(\mathbf{x}_t)\|^2$$

$$=\mathcal{O}\left(\frac{\sqrt{T}\left(\Delta(\theta,T,\delta)+\log^{(2\theta+1)}(\frac{1}{\delta})\log T\right)}{n}\times\left(d+\log(\frac{1}{\delta})\right)\right)$$

$$+\mathcal{O}\left(\frac{\Delta(\theta,T,\delta)+\log^{2\theta}(1/\delta)\log T}{\sqrt{T}}\right).$$

Taking $T \asymp \frac{n}{d}$, we have the following inequality with probability $1-2\delta$

$$\frac{1}{T}\sum_{t=1}^{T}\|\nabla F(\mathbf{x}_t)\|^2 = \mathcal{O}\left(\left(\frac{d}{n}\right)^{\frac{1}{2}}\left(\log(\frac{n}{d})\log^{(2\theta+2)}(\frac{1}{\delta})+\Delta(\theta,\frac{n}{d},\delta)\log(1/\delta)\right)\right),$$

which means with probability at least $1-\delta$ we have

$$\frac{1}{T}\sum_{t=1}^{T}\|\nabla F(\mathbf{x}_t)\|^2 =\mathcal{O}\left(\left(\frac{d}{n}\right)^{\frac{1}{2}}\left(\log(\frac{n}{d})\log^{(2\theta+2)}(\frac{1}{\delta})+\Delta(\theta,\frac{n}{d},\delta)\log(1/\delta)\right)\right)$$

$$=\mathcal{O}\left(\left(\frac{d}{n}\right)^{\frac{1}{2}}\left(\log(\frac{n}{d})\log^{(2\theta+2)}(\frac{1}{\delta})+\log^{\theta-1}(n/d\delta)\log^2(1/\delta)\mathbb{I}_{\theta>1}\right)\right).$$

The proof is complete. $\qquad\qquad\square$

### C.3 PROOF OF THEOREM 3.5

*Proof.* The proof of Theorem 3.5 is relatively complex and is divided into two parts.

**(1.)** In the first part, we prove the bound of $\|\mathbf{x}_{t+1}\|$, characterizing the bound of $B(\mathbf{0},R)$, i.e., at iterate $t+1$, $R = R_{t+1} = \|\mathbf{x}_{t+1}\|$. Recall that in (13), we need $\eta_t \le \frac{1}{8}\frac{(1-\gamma)^2}{\frac{L}{1-\gamma}+\frac{1}{2}L}$. Since $\eta_t = \frac{1}{\mu(S)(t+t_0)}$, when $t_0 \ge \frac{8(\frac{L}{1-\gamma}+\frac{1}{2}L)}{\mu(S)(1-\gamma)^2} = \frac{12L-4L\gamma}{\mu(S)(1-\gamma)^3}$, we have $\eta_t \le \frac{1}{8}\frac{(1-\gamma)^2}{\frac{L}{1-\gamma}+\frac{1}{2}L}$. Thus, we can use (23) to bound $\|\mathbf{x}_{t+1}\|$. According to (23), we have the following inequality with probability $1-\delta$ uniformly for all $t = 1,...T$

$$\|\mathbf{x}_{t+1}\| = \mathcal{O}\left(\log^{(\theta+\frac{1}{2})}(\frac{1}{\delta})(\sum_{t=1}^{T}\eta_t^2)^{\frac{1}{2}}+\left(\sum_{i=1}^{t}\eta_i\right)^{\frac{1}{2}}\left(\Delta^{\frac{1}{2}}(\theta,T,\delta)+\log^{\theta}(1/\delta)\left(\sum_{i=1}^{t}\eta_i^2\right)^{\frac{1}{2}}\right)\right)$$

$$\le \mathcal{O}\left(\left(\log^{(\theta+\frac{1}{2})}(\frac{1}{\delta})+\Delta^{\frac{1}{2}}(\theta,T,\delta)\right)\log^{\frac{1}{2}}T\right), \qquad\qquad (27)$$

where $\Delta(\theta,T,\delta) = \log^{\theta-1}(T/\delta)\log(1/\delta)\mathbb{I}_{\theta>1}$, and where the last inequality follows from $\eta_t = \frac{1}{\mu(S)(t+t_0)}$ with $t_0 \ge 1$ and Lemma B.5.

**(2.)** In the second part, we prove the bound of $F_S(\mathbf{x}_{T+1}) - F_S(\mathbf{x}(S))$. It is clear that

$$F_S(\mathbf{x}_{t+1}) - F_S(\mathbf{x}_t)$$

$$\le \langle\mathbf{x}_{t+1}-\mathbf{x}_t,\nabla F_S(\mathbf{x}_t)\rangle+\frac{1}{2}L\|\mathbf{x}_{t+1}-\mathbf{x}_t\|^2$$

$$\le -\gamma\langle\mathbf{m}_{t-1},\nabla F_S(\mathbf{x}_{t-1})\rangle+L\gamma\|\mathbf{m}_{t-1}\|^2-\langle\eta_t\nabla f(\mathbf{x}_t;z_{j_t}),\nabla F_S(\mathbf{x}_t)\rangle+\frac{1}{2}L\|\mathbf{m}_t\|^2$$

$$= -\gamma\langle\mathbf{m}_{t-1},\nabla F_S(\mathbf{x}_{t-1})\rangle+L\gamma\|\mathbf{m}_{t-1}\|^2-\langle\eta_t\nabla f(\mathbf{x}_t;z_{j_t})-\nabla F_S(\mathbf{x}_t),\nabla F_S(\mathbf{x}_t)\rangle$$

$$-\eta_t\|\nabla F_S(\mathbf{x}_t)\|^2+\frac{1}{2}L\|\mathbf{m}_t\|^2,$$

where the second inequality follows from (6). We can derive that

$$\frac{1}{2}\eta_t\|\nabla F_S(\mathbf{x}_t)\|^2 + F_S(\mathbf{x}_{t+1}) - F_S(\mathbf{x}_t)$$

$$\leq -\gamma\langle\mathbf{m}_{t-1}, \nabla F_S(\mathbf{x}_{t-1})\rangle + L\gamma\|\mathbf{m}_{t-1}\|^2 - \langle\eta_t\nabla f(\mathbf{x}_t; z_{j_t}) - \nabla F_S(\mathbf{x}_t), \nabla F_S(\mathbf{x}_t)\rangle$$

$$-\frac{1}{2}\eta_t\|\nabla F_S(\mathbf{x}_t)\|^2 + \frac{1}{2}L\|\mathbf{m}_t\|^2.$$

Since $\eta_t = \frac{1}{\mu(S)(t+t_0)}$, it implies that

$$\frac{1}{2}\eta_t\|\nabla F_S(\mathbf{x}_t)\|^2 + F_S(\mathbf{x}_{t+1}) - F_S(\mathbf{x}_S)$$

$$\leq (1 - \frac{2}{t+t_0})(F_S(\mathbf{x}_t) - F_S(\mathbf{x}_S)) - \gamma\langle\mathbf{m}_{t-1}, \nabla F_S(\mathbf{x}_{t-1})\rangle + L\gamma\|\mathbf{m}_{t-1}\|^2$$

$$- \langle\eta_t\nabla f(\mathbf{x}_t; z_{j_t}) - \nabla F_S(\mathbf{x}_t), \nabla F_S(\mathbf{x}_t)\rangle + \frac{1}{2}L\|\mathbf{m}_t\|^2.$$

Multiplying both sides by $(t + t_0)(t + t_0 - 1)$, we get

$$\frac{(t+t_0-1)}{2\mu(S)}\|\nabla F_S(\mathbf{x}_t)\|^2 + (t+t_0)(t+t_0-1)(F_S(\mathbf{x}_{t+1}) - F_S(\mathbf{x}_S))$$

$$\leq -(t+t_0)(t+t_0-1)\gamma\langle\mathbf{m}_{t-1}, \nabla F_S(\mathbf{x}_{t-1})\rangle + (t+t_0)(t+t_0-1)L\gamma\|\mathbf{m}_{t-1}\|^2$$

$$+ (t+t_0)(t+t_0-1)\frac{1}{2}L\|\mathbf{m}_t\|^2$$

$$+ (t+t_0-1)(t+t_0-2)(F_S(\mathbf{x}_t) - F_S(\mathbf{x}_S))$$

$$- (t+t_0)(t+t_0-1)\eta_t\langle\nabla f(\mathbf{x}_t; z_{j_t}) - \nabla F_S(\mathbf{x}_t), \nabla F_S(\mathbf{x}_t)\rangle.$$

Taking a summation from $t = 1$ to $t = T$, we derive that

$$\sum_{t=1}^{T}\frac{(t+t_0-1)}{2\mu(S)}\|\nabla F_S(\mathbf{x}_t)\|^2 + (T+t_0)(T+t_0-1)(F_S(\mathbf{x}_{T+1}) - F_S(\mathbf{x}_S))$$

$$\leq -\sum_{t=1}^{T}(t+t_0)(t+t_0-1)\gamma\langle\mathbf{m}_{t-1}, \nabla F_S(\mathbf{x}_{t-1})\rangle + \sum_{t=1}^{T}(t+t_0)(t+t_0-1)L\gamma\|\mathbf{m}_{t-1}\|^2$$

$$+ \sum_{t=1}^{T}(t+t_0)(t+t_0-1)\frac{1}{2}L\|\mathbf{m}_t\|^2$$

$$+ (t_0-1)(t_0-2)(F_S(\mathbf{x}_1) - F_S(\mathbf{x}_S))$$

$$- \sum_{t=1}^{T}(t+t_0)(t+t_0-1)\eta_t\langle\nabla f(\mathbf{x}_t; z_{j_t}) - \nabla F_S(\mathbf{x}_t), \nabla F_S(\mathbf{x}_t)\rangle.$$

Since $\mathbf{m}_0 = 0$, we get

$$\sum_{t=1}^{T}\frac{(t+t_0-1)}{2\mu(S)}\|\nabla F_S(\mathbf{x}_t)\|^2 + (T+t_0)(T+t_0-1)(F_S(\mathbf{x}_{T+1}) - F_S(\mathbf{x}_S))$$

$$\leq -\sum_{t=1}^{T}(t+t_0)(t+t_0-1)\gamma\langle\mathbf{m}_{t-1}, \nabla F_S(\mathbf{x}_{t-1})\rangle + \sum_{t=1}^{T-1}(t+t_0+1)(t+t_0)L\gamma\|\mathbf{m}_t\|^2$$

$$+ \sum_{t=1}^{T}(t+t_0)(t+t_0-1)\frac{1}{2}L\|\mathbf{m}_t\|^2$$

$$+ (t_0-1)(t_0-2)(F_S(\mathbf{x}_1) - F_S(\mathbf{x}_S))$$

$$- \sum_{t=1}^{T}(t+t_0)(t+t_0-1)\eta_t\langle\nabla f(\mathbf{x}_t; z_{j_t}) - \nabla F_S(\mathbf{x}_t), \nabla F_S(\mathbf{x}_t)\rangle. \tag{28}$$

We first bound the term $\sum_{t=1}^{T-1}(t + t_0 + 1)(t + t_0)\|\mathbf{m}_t\|^2$. Note that from the Jensen's inequality, we have

$$\|\mathbf{m}_t\|^2 = \|\gamma\mathbf{m}_{t-1} + \frac{1-\gamma}{1-\gamma}\eta_t\nabla f(\mathbf{x}_t; z_{j_t})\|^2 \leq \gamma\|\mathbf{m}_{t-1}\|^2 + \frac{1}{1-\gamma}\|\eta_t\nabla f(\mathbf{x}_t; z_{j_t})\|^2.$$

By recurrence, it gives that

$$\|\mathbf{m}_t\|^2 \leq \sum_{i=1}^{t}\frac{\gamma^{t-i}}{1-\gamma}\|\eta_i\nabla f(\mathbf{x}_i; z_{j_i})\|^2.$$

Thus, we have

$$\sum_{t=1}^{T-1}(t + t_0 + 1)(t + t_0)\|\mathbf{m}_t\|^2$$

$$\leq \sum_{t=1}^{T-1}(t + t_0 + 1)(t + t_0)\sum_{i=1}^{t}\frac{\gamma^{t-i}}{1-\gamma}\|\eta_i\nabla f(\mathbf{x}_i; z_{j_i})\|^2$$

$$= \sum_{t=1}^{T-1}\frac{\gamma^{-t}}{1-\gamma}\|\eta_t\nabla f(\mathbf{x}_t; z_{j_t})\|^2\sum_{i=t}^{T-1}\gamma^i(i + t_0 + 1)(i + t_0) \tag{29}$$

Considering $\sum_{i=t}^{T-1}(i + t_0 + 1)(i + t_0)\gamma^i$, we have

$$\sum_{i=t}^{T-1}(i + t_0 + 1)(i + t_0)\gamma^i$$

$$\leq \int_{t}^{T-1}(i + t_0 + 1)(i + t_0)\gamma^i di$$

$$\leq \int_{t}^{T-1}(i + t_0 + 1)^2\gamma^i di$$

$$= \frac{\gamma^i}{\ln\gamma}(i + t_0 + 1)^2\Big|_{i=t}^{i=T-1} - 2\int_{t}^{T-1}(i + t_0 + 1)\gamma^i di$$

$$= \frac{\gamma^i}{\ln\gamma}(i + t_0 + 1)^2\Big|_{i=t}^{i=T-1} - 2\Big[\frac{\gamma^i}{\ln^2\gamma}(i + t_0 + 1)\Big|_{i=t}^{i=T-1} - \int_{t}^{T-1}\gamma^i di\Big].$$

Solving the above integral, and since $\ln\gamma < 0$, we get

$$\sum_{i=t}^{T-1}(i + t_0 + 1)(i + t_0)\gamma^i$$

$$\leq -\frac{\gamma^t}{\ln\gamma}(t + t_0 + 1)^2 + 2\frac{\gamma^t}{\ln^2\gamma}(t + t_0 + 1) - 2\frac{\gamma^t}{\ln\gamma} \leq (C_\gamma)\gamma^t(t + t_0 + 1)^2, \tag{30}$$

where $C_\gamma = 1 + 2\frac{1}{\ln^2\gamma} - \frac{3}{\ln\gamma}$, which is a constant only depend on $\gamma$. Thus, according to (29), we have

$$\sum_{t=1}^{T-1}(t + t_0 + 1)(t + t_0)\|\mathbf{m}_t\|^2 \leq \sum_{t=1}^{T-1}(t + t_0 + 1)^2\frac{(C_\gamma)}{(1-\gamma)}\|\eta_t\nabla f(\mathbf{x}_t; z_{j_t})\|^2$$

$$\leq \frac{(C_\gamma)}{(1-\gamma)\mu(S)^2}\sum_{t=1}^{T-1}\frac{(t + t_0 + 1)^2}{(t + t_0)^2}\|\nabla f(\mathbf{x}_t; z_{j_t})\|^2.$$

And since $\frac{(t+t_0+1)^2}{(t+t_0)^2} = (1 + \frac{1}{t+t_0})^2 \le 4$, then we have

$$\sum_{t=1}^{T-1}(t + t_0 + 1)(t + t_0)\|\mathbf{m}_t\|^2$$

$$\le \frac{(4C_\gamma)}{(1-\gamma)\mu(S)^2}\sum_{t=1}^{T-1}\|\nabla f(\mathbf{x}_t; z_{j_t})\|^2$$

$$\le \frac{(8C_\gamma)}{(1-\gamma)\mu(S)^2}\Big(\sum_{t=1}^{T-1}\|\nabla f(\mathbf{x}_t; z_{j_t}) - \nabla F_S(\mathbf{x}_t)\|^2 + \|\nabla F_S(\mathbf{x}_t)\|^2\Big).$$

Since $\|\nabla f(\mathbf{x}_t; z_{j_t}) - \nabla F_S(\mathbf{x}_t)\| \sim subW(\theta, K)$, we get $\mathbb{E}\left[\exp\left(\frac{\|\nabla f(\mathbf{x}_t; z_{j_t}) - \nabla F_S(\mathbf{x}_t)\|^2}{K^2}\right)^{\frac{1}{2\theta}}\right] \le 2$.
According to Lemma B.2, we get the following inequality with probability at least $1 - \delta$

$$\sum_{t=1}^{T-1}\|\nabla f(\mathbf{x}_t; z_{j_t}) - \nabla F_S(\mathbf{x}_t)\|^2 \le (T-1)K^2 g(2\theta)\log^{2\theta}(2/\delta).$$

Thus, with probability at least $1 - \delta$, we have

$$\sum_{t=1}^{T-1}(t + t_0 + 1)(t + t_0)\|\mathbf{m}_t\|^2 \le \frac{(8C_\gamma)}{(1-\gamma)\mu(S)^2}(T-1)K^2 g(2\theta)\log^{2\theta}(2/\delta)$$

$$+ \sum_{t=1}^{T-1}\frac{(8C_\gamma)}{(1-\gamma)\mu(S)^2}\|\nabla F_S(\mathbf{x}_t)\|^2. \tag{31}$$

Similarly, with probability at least $1 - \delta$, we can derive

$$\sum_{t=1}^{T}(t + t_0)(t + t_0 - 1)\|\mathbf{m}_t\|^2$$

$$\le \frac{(8C_\gamma)}{(1-\gamma)\mu(S)^2}TK^2 g(2\theta)\log^{2\theta}(2/\delta) + \sum_{t=1}^{T}\frac{(8C_\gamma)}{(1-\gamma)\mu(S)^2}\|\nabla F_S(\mathbf{x}_t)\|^2.$$

We then bound $-\sum_{t=1}^{T}(t + t_0)(t + t_0 - 1)\langle\mathbf{m}_{t-1}, \nabla F_S(\mathbf{x}_{t-1})\rangle$. Recall that from (7), we know

$$-\langle\mathbf{m}_t, \nabla F_S(\mathbf{x}_t)\rangle \le L\sum_{i=1}^{t-1}\gamma^{t-i}\|\mathbf{m}_i\|^2 - \sum_{i=1}^{t}\gamma^{t-i}\langle\eta_i\nabla f(\mathbf{x}_i; z_{j_i}), \nabla F_S(\mathbf{x}_i)\rangle.$$

Since $\mathbf{m}_0 = 0$, we have

$$-\sum_{t=1}^{T}(t+t_0)(t+t_0-1)\langle\mathbf{m}_{t-1}, \nabla F_S(\mathbf{x}_{t-1})\rangle$$

$$= -\sum_{t=1}^{T-1}(t+t_0+1)(t+t_0)\langle\mathbf{m}_t, \nabla F_S(\mathbf{x}_t)\rangle$$

$$\leq \sum_{t=1}^{T-1}(t+t_0+1)(t+t_0)L\sum_{i=1}^{t-1}\gamma^{t-i}\|\mathbf{m}_i\|^2$$

$$-\sum_{t=1}^{T-1}(t+t_0+1)(t+t_0)\sum_{i=1}^{t}\gamma^{t-i}\langle\eta_i\nabla f(\mathbf{x}_i; z_{j_i}), \nabla F_S(\mathbf{x}_i)\rangle$$

$$\leq \sum_{t=1}^{T-1}(t+t_0+1)(t+t_0)L\sum_{i=1}^{t}\gamma^{t-i}\|\mathbf{m}_i\|^2$$

$$-\sum_{t=1}^{T-1}(t+t_0+1)(t+t_0)\sum_{i=1}^{t}\gamma^{t-i}\langle\eta_i\nabla f(\mathbf{x}_i; z_{j_i}), \nabla F_S(\mathbf{x}_i)\rangle$$

$$= \sum_{t=1}^{T-1}\gamma^{-t}\|\mathbf{m}_t\|^2L\sum_{i=t}^{T-1}\gamma^i(i+t_0+1)(i+t_0)$$

$$-\sum_{t=1}^{T-1}\gamma^{-t}\langle\eta_t\nabla f(\mathbf{x}_t; z_{j_t}), \nabla F_S(\mathbf{x}_t)\rangle\sum_{i=t}^{T-1}(i+t_0+1)(i+t_0)\gamma^i$$

$$= \sum_{t=1}^{T-1}\gamma^{-t}\|\mathbf{m}_t\|^2L\sum_{i=t}^{T-1}\gamma^i(i+t_0+1)(i+t_0)$$

$$-\sum_{t=1}^{T-1}\gamma^{-t}\langle\eta_t(\nabla f(\mathbf{x}_t; z_{j_t}) - \nabla F_S(\mathbf{x}_t)), \nabla F_S(\mathbf{x}_t)\rangle\sum_{i=t}^{T-1}(i+t_0+1)(i+t_0)\gamma^i$$

$$-\sum_{t=1}^{T-1}\gamma^{-t}\langle\eta_t\nabla F_S(\mathbf{x}_t), \nabla F_S(\mathbf{x}_t)\rangle\sum_{i=t}^{T-1}(i+t_0+1)(i+t_0)\gamma^i,$$

where the second equation holds by using Lemma B.6.

With a similar analysis to (31), it is clear that with probability $1 - \delta$

$$\sum_{t=1}^{T-1}\gamma^{-t}\|\mathbf{m}_t\|^2L\sum_{i=t}^{T-1}\gamma^i(i+t_0+1)(i+t_0) \leq LC_\gamma\sum_{t=1}^{T-1}\|\mathbf{m}_t\|^2(t+t_0+1)^2$$

$$\leq L(C_\gamma)\frac{(8C_\gamma)}{(1-\gamma)\mu(S)^2}(T-1)K^2g(2\theta)\log^{2\theta}(2/\delta) + \sum_{t=1}^{T-1}L(C_\gamma)\frac{(8C_\gamma)}{(1-\gamma)\mu(S)^2}\|\nabla F_S(\mathbf{x}_t)\|^2.$$

And we also have

$$-\sum_{t=1}^{T-1}\gamma^{-t}\langle\eta_t\nabla F_S(\mathbf{x}_t), \nabla F_S(\mathbf{x}_t)\rangle\sum_{i=t}^{T-1}(i+t_0+1)(i+t_0)\gamma^i$$

$$\leq -\sum_{t=1}^{T-1}\gamma^{-t}(t+t_0+1)(t+t_0)\langle\eta_t\nabla F_S(\mathbf{x}_t), \nabla F_S(\mathbf{x}_t)\rangle\sum_{i=t}^{T-1}\gamma^i$$

$$\leq -\sum_{t=1}^{T-1}(t+t_0+1)(t+t_0)\langle\eta_t\nabla F_S(\mathbf{x}_t), \nabla F_S(\mathbf{x}_t)\rangle$$

$$= -\sum_{t=1}^{T-1}(t+t_0+1)(t+t_0)\eta_t\|\nabla F_S(\mathbf{x}_t)\|^2.$$

Thus, we have

$$-\sum_{t=1}^{T}(t+t_0)(t+t_0-1)\langle \mathbf{m}_{t-1}, \nabla F_S(\mathbf{x}_{t-1})\rangle$$

$$\leq -\sum_{t=1}^{T-1}\gamma^{-t}\langle \eta_t(\nabla f(\mathbf{x}_t; z_{j_t}) - \nabla F_S(\mathbf{x}_t)), \nabla F_S(\mathbf{x}_t)\rangle \sum_{i=t}^{T-1}(i+t_0+1)(i+t_0)\gamma^i$$

$$-\sum_{t=1}^{T-1}(t+t_0+1)(t+t_0)\eta_t\|\nabla F_S(\mathbf{x}_t)\|^2 + L(C_\gamma)\frac{(8C_\gamma)}{(1-\gamma)\mu(S)^2}(T-1)K^2 g(2\theta)\log^{2\theta}(2/\delta)$$

$$+\sum_{t=1}^{T-1}L(C_\gamma)\frac{(8C_\gamma)}{(1-\gamma)\mu(S)^2}\|\nabla F_S(\mathbf{x}_t)\|^2.$$

We now consider the term $-\sum_{t=1}^{T-1}\gamma^{-t}\langle \eta_t(\nabla f(\mathbf{x}_t; z_{j_t}) - \nabla F_S(\mathbf{x}_t)), \nabla F_S(\mathbf{x}_t)\rangle \sum_{i=t}^{T-1}(i+t_0+1)(i+t_0)\gamma^i$. Denoted by $\xi_t = -\gamma^{-t}\langle \eta_t(\nabla f(\mathbf{x}_t; z_{j_t}) - \nabla F_S(\mathbf{x}_t)), \nabla F_S(\mathbf{x}_t)\rangle \sum_{i=t}^{T-1}(i+t_0+1)(i+t_0)\gamma^i$. We know that $\mathbb{E}_{j_t}\xi_t = -\mathbb{E}_{j_t}\gamma^{-t}\langle \eta_t(\nabla f(\mathbf{x}_t; z_{j_t}) - \nabla F_S(\mathbf{x}_t)), \nabla F_S(\mathbf{x}_t)\rangle \sum_{i=t}^{T-1}(i+t_0+1)(i+t_0)\gamma^i = 0$, implying that it is a martingale difference sequence. We use Lemma B.4 to bound this term.

From (30), it is clear that $|\gamma^{-t}\langle \eta_t(\nabla f(\mathbf{x}_t; z_{j_t}) - \nabla F_S(\mathbf{x}_t)), \nabla F_S(\mathbf{x}_t)\rangle \sum_{i=t}^{T-1}(i+t_0+1)(i+t_0)\gamma^i| \leq (C_\gamma)(t+t_0+1)^2\eta_t\|\nabla f(\mathbf{x}_t; z_{j_t}) - \nabla F_S(\mathbf{x}_t))\|\|\nabla F_S(\mathbf{x}_t)\|$. We set

$$K_{t-1} = C_\gamma(t+t_0+1)^2\eta_t K\|\nabla F_S(\mathbf{x}_t)\| = C_\gamma(t+t_0+1)^2\frac{1}{\mu(S)(t+t_0)}K\|\nabla F_S(\mathbf{x}_t)\|.$$

We also set $\beta = 0$, $\lambda = \frac{1}{2\alpha}$, and $x = 2\alpha\log(1/\delta)$. For brevity, we denote $\Xi = 2C_\gamma(t+t_0+1)\mu(S)^{-1}K$ and $\Xi_T = 2C_\gamma(T+t_0+1)\mu(S)^{-1}K$. Moreover, according to the smoothness assumption, we know $\|\nabla F_S(\mathbf{x}_t)\| \leq (L\|\mathbf{x}_t\| + B)$.

If $\theta = \frac{1}{2}$, for all $\alpha > 0$, we have the following inequality with probability $1-\delta$

$$-\sum_{t=1}^{T-1}\gamma^{-t}\langle \eta_t\nabla f(\mathbf{x}_t; z_{j_t}), \nabla F_S(\mathbf{x}_t)\rangle \sum_{i=t}^{T-1}(i+t_0+1)(i+t_0)\gamma^i$$

$$\leq 2\alpha\log(1/\delta) + \frac{a}{\alpha}\sum_{t=1}^{T-1}\Xi^2\|\nabla F_S(\mathbf{x}_t)\|^2.$$

If $\frac{1}{2} < \theta \leq 1$, we set $m_t = \Xi(L\|\mathbf{x}_t\| + B)$. Then for all $\alpha \geq b\Xi_T(L\|\mathbf{x}_T\| + B)$, we have the following inequality with probability $1-\delta$

$$-\sum_{t=1}^{T-1}\gamma^{-t}\langle \eta_t\nabla f(\mathbf{x}_t; z_{j_t}), \nabla F_S(\mathbf{x}_t)\rangle \sum_{i=t}^{T-1}(i+t_0+1)(i+t_0)\gamma^i$$

$$\leq 2\alpha\log(1/\delta) + \frac{a}{\alpha}\sum_{t=1}^{T-1}\Xi^2\|\nabla F_S(\mathbf{x}_t)\|^2.$$

If $\theta > 1$, we set $m_t = \Xi(L\|\mathbf{x}_t\| + B)$ and $\delta = \delta$. Then, for all $\alpha \geq b\Xi_T(L\|\mathbf{x}_T\| + B)$, we have the following inequality with probability $1-3\delta$

$$-\sum_{t=1}^{T-1}\gamma^{-t}\langle \eta_t\nabla f(\mathbf{x}_t; z_{j_t}), \nabla F_S(\mathbf{x}_t)\rangle \sum_{i=t}^{T-1}(i+t_0+1)(i+t_0)\gamma^i$$

$$\leq 2\alpha\log(1/\delta) + \frac{a}{\alpha}\sum_{t=1}^{T-1}\Xi^2\|\nabla F_S(\mathbf{x}_t)\|^2.$$

We now consider the last term $-(t + t_0)(t + t_0 - 1)\eta_t\langle\nabla f(\mathbf{x}_t; z_{j_t}) - \nabla F_S(\mathbf{x}_t), \nabla F_S(\mathbf{x}_t)\rangle$. With a similar analysis, we set $\xi_t = -(t + t_0)(t + t_0 - 1)\eta_t\langle\nabla f(\mathbf{x}_t; z_{j_t}) - \nabla F_S(\mathbf{x}_t), \nabla F_S(\mathbf{x}_t)\rangle$ and

$$K_{t-1} = (t + t_0)(t + t_0 - 1)\eta_t K\|\nabla F_S(\mathbf{x}_t)\| = \mu(S)^{-1}(t + t_0 - 1)K\|\nabla F_S(\mathbf{x}_t)\|.$$

We also set $\beta = 0$, $\lambda = \frac{1}{2\alpha}$, and $x = 2\alpha\log(1/\delta)$. According to the smoothness assumption, we know $\|\nabla F_S(\mathbf{x}_t)\| \leq (L\|\mathbf{x}_t\| + B)$.

If $\theta = \frac{1}{2}$, for all $\alpha > 0$, we have the following inequality with probability at least $1 - \delta$

$$-\sum_{t=1}^{T}(t + t_0)(t + t_0 - 1)\eta_t\langle\nabla f(\mathbf{x}_t; z_{j_t}) - \nabla F_S(\mathbf{x}_t), \nabla F_S(\mathbf{x}_t)\rangle$$

$$\leq 2\alpha\log(1/\delta) + \frac{aK^2}{\mu(S)^2\alpha}\sum_{t=1}^{T}(t + t_0 - 1)^2\|\nabla F_S(\mathbf{x}_t)\|^2.$$

If $\frac{1}{2} < \theta \leq 1$, we set $m_t = \mu(S)^{-1}(t + t_0 - 1)K(L\|\mathbf{x}_t\| + B)$. Then for all $\alpha \geq b\mu(S)^{-1}(T + t_0 - 1)K(L\|\mathbf{x}_T\| + B)$, we have the following inequality with probability at least $1 - \delta$

$$-\sum_{t=1}^{T}(t + t_0)(t + t_0 - 1)\eta_t\langle\nabla f(\mathbf{x}_t; z_{j_t}) - \nabla F_S(\mathbf{x}_t), \nabla F_S(\mathbf{x}_t)\rangle$$

$$\leq 2\alpha\log(1/\delta) + \frac{aK^2}{\mu(S)^2\alpha}\sum_{t=1}^{T}(t + t_0 - 1)^2\|\nabla F_S(\mathbf{x}_t)\|^2.$$

If $\theta > 1$, we set $m_t = \mu(S)^{-1}(t + t_0 - 1)K(L\|\mathbf{x}_t\| + B)$ and $\delta = \delta$. Then, for all $\alpha \geq b\mu(S)^{-1}(T + t_0 - 1)K(L\|\mathbf{x}_T\| + B)$, we have the following inequality with probability at least $1 - 3\delta$

$$-\sum_{t=1}^{T}(t + t_0)(t + t_0 - 1)\eta_t\langle\nabla f(\mathbf{x}_t; z_{j_t}) - \nabla F_S(\mathbf{x}_t), \nabla F_S(\mathbf{x}_t)\rangle$$

$$\leq 2\alpha\log(1/\delta) + \frac{aK^2}{\mu(S)^2\alpha}\sum_{t=1}^{T}(t + t_0 - 1)^2\|\nabla F_S(\mathbf{x}_t)\|^2.$$

Finally, combining with these terms, we derive

$$\sum_{t=1}^{T}\frac{(t + t_0 - 1)}{2\mu(S)}\|\nabla F_S(\mathbf{x}_t)\|^2 - \frac{aK^2}{\mu(S)^2\alpha}\sum_{t=1}^{T}(t + t_0 - 1)^2\|\nabla F_S(\mathbf{x}_t)\|^2$$

$$- \frac{L}{2}\sum_{t=1}^{T}\frac{(8C_\gamma)}{(1 - \gamma)\mu(S)^2}\|\nabla F_S(\mathbf{x}_t)\|^2$$

$$- L\gamma\sum_{t=1}^{T-1}\frac{(8C_\gamma)}{(1 - \gamma)\mu(S)^2}\|\nabla F_S(\mathbf{x}_t)\|^2 + \sum_{t=1}^{T-1}(t + t_0 + 1)(t + t_0)\eta_t\|\nabla F_S(\mathbf{x}_t)\|^2$$

$$- \sum_{t=1}^{T-1}L\gamma(C_\gamma)\frac{(8C_\gamma)}{(1 - \gamma)\mu(S)^2}\|\nabla F_S(\mathbf{x}_t)\|^2$$

$$- \gamma\frac{a}{\alpha}\sum_{t=1}^{T-1}\Xi^2\|\nabla F_S(\mathbf{x}_t)\|^2 + (T + t_0)(T + t_0 - 1)(F_S(\mathbf{x}_{T+1}) - F_S(\mathbf{x}(S)))$$

$$\leq L\gamma\frac{(8C_\gamma)}{(1 - \gamma)\mu(S)^2}(T - 1)K^2 g(2\theta)\log^{2\theta}(2/\delta) + \frac{L}{2}\frac{(8C_\gamma)}{(1 - \gamma)\mu(S)^2}TK^2 g(2\theta)\log^{2\theta}(2/\delta)$$

$$+ L\gamma(C_\gamma)\frac{(8C_\gamma)}{(1 - \gamma)\mu(S)^2}(T - 1)K^2 g(2\theta)\log^{2\theta}(2/\delta) + (t_0 - 1)(t_0 - 2)(F_S(\mathbf{x}_1) - F_S(\mathbf{x}(S)))$$

$$+ 2\alpha\log(1/\delta) + \gamma 2\alpha\log(1/\delta). \tag{32}$$

We want

$$\frac{(t+t_0-1)}{2\mu(S)} - \frac{aK^2}{\mu(S)^2\alpha}(t+t_0-1)^2 - \frac{L}{2}\frac{(8C_\gamma)}{(1-\gamma)^2\mu(S)^2} \geq 0$$

and

$$\frac{(t+t_0+1)}{\mu(S)} - L\gamma\frac{(8C_\gamma)}{(1-\gamma)\mu(S)^2} - L\gamma(C_\gamma)\frac{(8C_\gamma)}{(1-\gamma)\mu(S)^2} - \gamma\frac{a}{\alpha}\Xi^2 \geq 0.$$

Thus, we assume that $t_0$ satisfies the following conditions

$$\frac{(t_0-1)}{2\mu(S)} \geq \frac{L}{2}\frac{(8C_\gamma)}{(1-\gamma)^2\mu(S)^2};$$

and

$$\frac{(t_0+1)}{\mu(S)} \geq L\gamma\frac{(8C_\gamma)}{(1-\gamma)\mu(S)^2} + L\gamma(C_\gamma)\frac{(8C_\gamma)}{(1-\gamma)\mu(S)^2},$$

which means that

$$t_0 \geq \frac{(8C_\gamma)L}{(1-\gamma)^2\mu(S)} + 1;$$

and

$$t_0 \geq \frac{8C_\gamma(L\gamma + L\gamma(C_\gamma))}{(1-\gamma)\mu(S)} - 1.$$

Thus, we can further derive that $\alpha \geq \frac{aK^2(t+t_0-1)^2}{\frac{(t+t_0-1)}{2\mu(S)} - \frac{L}{2}\frac{(8C_\gamma)}{(1-\gamma)^2\mu(S)^2}}$ and

$$\alpha \geq \frac{\gamma a(2C_\gamma(t+t_0+1)\mu(S)^{-1}K)^2}{\frac{(t+t_0+1)}{\mu(S)} - L\gamma\frac{(8C_\gamma)}{(1-\gamma)^2\mu(S)^2} - L\gamma(C_\gamma)\frac{(8C_\gamma)}{(1-\gamma)^2\mu(S)^2}}.$$

When $\theta = \frac{1}{2}$, the above lower bounds of $\alpha$ are: $\alpha \geq \frac{aK^2(t+t_0-1)^2}{\frac{(t+t_0-1)}{2\mu(S)} - \frac{L}{2}\frac{(8C_\gamma)}{(1-\gamma)^2\mu(S)^2}}$,

$\alpha \geq \frac{\gamma a(2C_\gamma(t+t_0+1)\mu(S)^{-1}K)^2}{\frac{(t+t_0+1)}{\mu(S)} - L\gamma\frac{(8C_\gamma)}{(1-\gamma)^2\mu(S)^2} - L\gamma(C_\gamma)\frac{(8C_\gamma)}{(1-\gamma)^2\mu(S)^2}}$, and $\alpha > 0$, which implies that we should choose $\alpha = \mathcal{O}(T)$.

When $\frac{1}{2} < \theta \leq 1$, the above lower bounds of $\alpha$ are: $\alpha \geq \frac{aK^2(t+t_0-1)^2}{\frac{(t+t_0-1)}{2\mu(S)} - \frac{L}{2}\frac{(8C_\gamma)}{(1-\gamma)^2\mu(S)^2}}$, $\alpha \geq$

$\frac{\gamma a(2C_\gamma(t+t_0+1)\mu(S)^{-1}K)^2}{\frac{(t+t_0+1)}{\mu(S)} - L\gamma\frac{(8C_\gamma)}{(1-\gamma)^2\mu(S)^2} - L\gamma(C_\gamma)\frac{(8C_\gamma)}{(1-\gamma)^2\mu(S)^2}}$, $\alpha \geq b\Xi_T(L\|\mathbf{x}_T\| + B)$, and $\alpha \geq b\mu(S)^{-1}(T + t_0 - 1)K(L\|\mathbf{x}_T\| + B)$, which implies that we should choose $\alpha = \mathcal{O}\left(T\log^{(\theta+\frac{1}{2})}(\frac{1}{\delta})\log^{\frac{1}{2}}T\right)$.

When $\theta > 1$, the above lower bounds of $\alpha$ are: $\alpha \geq \frac{aK^2(t+t_0-1)^2}{\frac{(t+t_0-1)}{2\mu(S)} - \frac{L}{2}\frac{(8C_\gamma)}{(1-\gamma)^2\mu(S)^2}}$,

$\alpha \geq \frac{\gamma a(2C_\gamma(t+t_0+1)\mu(S)^{-1}K)^2}{\frac{(t+t_0+1)}{\mu(S)} - L\gamma\frac{(8C_\gamma)}{(1-\gamma)^2\mu(S)^2} - L\gamma(C_\gamma)\frac{(8C_\gamma)}{(1-\gamma)^2\mu(S)^2}}$, $\alpha \geq b\Xi_T(L\|\mathbf{x}_T\| + B)$, and $\alpha \geq b\mu(S)^{-1}(T + t_0 - 1)K(L\|\mathbf{x}_T\| + B)$, which implies that we should choose

$$\alpha = \mathcal{O}\left(\log^{\theta-1}(\frac{T}{\delta})T\left(\log^{(\theta+\frac{1}{2})}(\frac{1}{\delta}) + \log^{\frac{\theta-1}{2}}(T/\delta)\log^{\frac{1}{2}}(1/\delta)\right)\log^{\frac{1}{2}}T\right).$$

Note that the bound of $\|\mathbf{x}_T\|$ comes from (27).

Thus, we derive that

$$(T+t_0)(T+t_0-1)(F_S(\mathbf{x}_{t+1}) - F_S(\mathbf{x}(S)))$$

$$\leq L\gamma\frac{(8C_\gamma)}{(1-\gamma)\mu(S)^2}(T-1)K^2g(2\theta)\log^{2\theta}(2/\delta) + \frac{L}{2}\frac{(8C_\gamma)}{(1-\gamma)\mu(S)^2}TK^2g(2\theta)\log^{2\theta}(2/\delta)$$

$$+ L\gamma(C_\gamma)\frac{(8C_\gamma)}{(1-\gamma)\mu(S)^2}(T-1)K^2g(2\theta)\log^{2\theta}(2/\delta)$$

$$+ (t_0-1)(t_0-2)(F_S(\mathbf{x}_1) - F_S(\mathbf{x}(S))) + 2\alpha\log(1/\delta) + \gamma2\alpha\log(1/\delta).$$

Putting the previous bounds together.

If $\theta = 1$, with probability $1 - 6\delta$, we have

$$F_S(\mathbf{x}_{T+1}) - F_S(\mathbf{x}(S)) = \mathcal{O}\left(\frac{\log(1/\delta)}{T}\right).$$

If $\frac{1}{2} < \theta \leq 1$, with probability $1 - 7\delta$, we have

$$F_S(\mathbf{x}_{T+1}) - F_S(\mathbf{x}(S)) = \mathcal{O}\left(\frac{\log^{(\theta+\frac{1}{2})}(\frac{1}{\delta})\log^{\frac{1}{2}}T}{T}\log(\frac{1}{\delta})\right).$$

If $\theta > 1$, with probability $1 - 10\delta$, we have

$$F_S(\mathbf{x}_{T+1}) - F_S(\mathbf{x}(S)) = \mathcal{O}\left(\frac{\left(\log^{(\theta+\frac{1}{2})}(\frac{1}{\delta}) + \Delta^{\frac{1}{2}}(\theta,T,\delta)\right)\log^{\frac{1}{2}}T}{T}\log^{\theta-1}(\frac{T}{\delta})\log(\frac{1}{\delta})\right).$$

The above bounds mean that with probability $1 - \delta$, there holds

$$F_S(\mathbf{x}_{T+1}) - F_S(\mathbf{x}(S)) = \begin{cases} \mathcal{O}\left(\frac{\log(1/\delta)}{T}\right) & \text{if} \quad \theta = \frac{1}{2}, \\ \mathcal{O}\left(\frac{\log^{(\theta+\frac{3}{2})}(\frac{1}{\delta})\log^{\frac{1}{2}}T}{T}\right) & \text{if} \quad \theta \in (\frac{1}{2}, 1], \\ \mathcal{O}\left(\frac{\log^{(\theta+\frac{3}{2})}(\frac{1}{\delta})\log^{\frac{3(\theta-1)}{2}}(T/\delta)\log^{\frac{1}{2}}T}{T}\right) & \text{if} \quad \theta > 1. \end{cases} \tag{33}$$

The proof is complete. $\qquad\square$

### C.4 Proof of Theorem 3.7

*Proof.* According to Assumption 2.6, we know

$$F(\mathbf{x}_{T+1}) - F(\mathbf{x}^*) \leq \frac{1}{4\mu}\|\nabla F(\mathbf{x}_{T+1})\|^2 \leq \frac{1}{2\mu}(\|\nabla F(\mathbf{x}_{T+1}) - \nabla F_S(\mathbf{x}_{T+1})\|^2 + \|\nabla F_S(\mathbf{x}_{T+1})\|^2). \tag{34}$$

Furthermore, from (27) and Lemma B.8, with probability $1 - \delta$ we have

$$\|\nabla F(\mathbf{x}_{T+1}) - \nabla F_S(\mathbf{x}_{T+1})\|^2 = \mathcal{O}\left(\frac{d + \log(\frac{1}{\delta})}{n}\|\mathbf{x}_{T+1}\|^2\right)$$

$$= \mathcal{O}\left(\frac{d + \log(\frac{1}{\delta})}{n}\left(\log^{(2\theta+1)}(\frac{1}{\delta}) + \Delta(\theta,T,\delta)\right)\log T\right). \tag{35}$$

From the smoothness property in Lemma B.7 and the convergence bound in (33), with probability $1 - \delta$, there holds

$$\|\nabla F_S(\mathbf{x}_{T+1})\|^2 \leq (2L)(F_S(\mathbf{x}_{T+1}) - F_S(\mathbf{x}(S)))$$

$$= \begin{cases} \mathcal{O}\left(\frac{\log(1/\delta)}{T}\right) & \text{if} \quad \theta = \frac{1}{2}, \\ \mathcal{O}\left(\frac{\log^{(\theta+\frac{3}{2})}(\frac{1}{\delta})\log^{\frac{1}{2}}T}{T}\right) & \text{if} \quad \theta \in (\frac{1}{2}, 1], \\ \mathcal{O}\left(\frac{\log^{(\theta+\frac{3}{2})}(\frac{1}{\delta})\log^{\frac{3(\theta-1)}{2}}(T/\delta)\log^{\frac{1}{2}}T}{T}\right) & \text{if} \quad \theta > 1. \end{cases} \tag{36}$$

Plugging (35) and (36) into (34), we derive that with probability $1 - 2\delta$, there holds: (1.) if $\theta = \frac{1}{2}$,

$$F(\mathbf{x}_{T+1}) - F(\mathbf{x}^*) = \mathcal{O}\left(\frac{\log(1/\delta)}{T} + \frac{d + \log(\frac{1}{\delta})}{n}\log^2(\frac{1}{\delta})\log T\right);$$

(2.) if $\theta \in (\frac{1}{2}, 1]$,

$$F(\mathbf{x}_{T+1}) - F(\mathbf{x}^*) = \mathcal{O}\left(\frac{\log^{(\theta+\frac{3}{2})}(\frac{1}{\delta})\log^{\frac{1}{2}} T}{T} + \frac{d + \log(\frac{1}{\delta})}{n}\log^{(2\theta+1)}(\frac{1}{\delta})\log T\right);$$

(3.) if $\theta > 1$,

$$F(\mathbf{x}_{T+1}) - F(\mathbf{x}^*)$$
$$= \mathcal{O}\left(\frac{\log^{(\theta+\frac{3}{2})}(\frac{1}{\delta})\log^{\frac{3(\theta-1)}{2}}(T/\delta)\log^{\frac{1}{2}} T}{T} + \frac{d + \log(\frac{1}{\delta})}{n}\left(\log^{(2\theta+1)}(\frac{1}{\delta}) + \Delta(\theta, T, \delta)\right)\log T\right).$$

We choose $T \asymp n$, then we get with probability at least $1 - \delta$, there holds

$$F(\mathbf{x}_{T+1}) - F(\mathbf{x}^*) = \begin{cases} \mathcal{O}\left(\frac{d+\log(\frac{1}{\delta})}{n}\log^2(\frac{1}{\delta})\log n\right) & \text{if} \quad \theta = \frac{1}{2}, \\ \mathcal{O}\left(\frac{d+\log(\frac{1}{\delta})}{n}\log^{(2\theta+1)}(\frac{1}{\delta})\log n\right) & \text{if} \quad \theta \in (\frac{1}{2}, 1], \\ \mathcal{O}\left(\frac{d+\log(\frac{1}{\delta})}{n}\log^{(2\theta+1)}(\frac{1}{\delta})\log^{\frac{3(\theta-1)}{2}}(\frac{n}{\delta})\log n\right) & \text{if} \quad \theta > 1. \end{cases}$$

The proof is complete. $\qquad\square$

## C.5 PROOF OF THEOREM 3.9

*Proof.* From Lemma B.9, with probability $1 - \delta$ we have

$$\|\nabla F(\mathbf{w}_{T+1}) - \nabla F_S(\mathbf{w}_{T+1})\|^2$$

$$\leq \left(\|\nabla F_S(\mathbf{w}_{T+1})\| + \frac{\mu}{n} + 2\frac{B_* \log(4/\delta)}{n} + 2\sqrt{\frac{2\mathbb{E}[\|\nabla f(\mathbf{x}^*; z)\|^2]\log(4/\delta)}{n}}\right)^2$$

$$\leq 4\left(\|\nabla F_S(\mathbf{w}_{T+1})\|^2 + 4\frac{B_*^2\log^2(4/\delta)}{n^2} + 8\frac{\mathbb{E}[\|\nabla f(\mathbf{x}^*; z)\|^2]\log(4/\delta)}{n} + \frac{\mu^2}{n^2}\right).$$

From the smoothness property in Lemma B.7, if $f$ is nonnegative and $L$-smooth, we have $\|\nabla f(\mathbf{x}^*; z)\|^2 \leq 2L\nabla f(\mathbf{x}^*; z)$, implying that $\mathbb{E}[\|\nabla f(\mathbf{x}^*; z)\|^2] \leq 2LF(\mathbf{x}^*)$. Thus, with probability $1 - \delta$ we have

$$\|\nabla F(\mathbf{w}_{T+1}) - \nabla F_S(\mathbf{w}_{T+1})\|^2$$

$$\leq 4\left(\|\nabla F_S(\mathbf{w}_{T+1})\|^2 + 4\frac{B_*^2\log^2(4/\delta)}{n^2} + \frac{16LF(\mathbf{x}^*)\log(4/\delta)}{n} + \frac{\mu^2}{n^2}\right). \qquad (37)$$

Again, from the smoothness property in Lemma B.7 and the convergence bound in (33), with probability $1 - \delta$, there holds

$$\|\nabla F_S(\mathbf{x}_{T+1})\|^2 \leq (2L)(F_S(\mathbf{x}_{T+1}) - F_S(\mathbf{x}(S)))$$

$$= \begin{cases} \mathcal{O}\left(\frac{\log(1/\delta)}{T}\right) & \text{if} \quad \theta = \frac{1}{2}, \\ \mathcal{O}\left(\frac{\log^{(\theta+\frac{3}{2})}(\frac{1}{\delta})\log^{\frac{1}{2}} T}{T}\right) & \text{if} \quad \theta \in (\frac{1}{2}, 1], \\ \mathcal{O}\left(\frac{\log^{(\theta+\frac{3}{2})}(\frac{1}{\delta})\log^{\frac{3(\theta-1)}{2}}(T/\delta)\log^{\frac{1}{2}} T}{T}\right) & \text{if} \quad \theta > 1. \end{cases} \qquad (38)$$

Plugging (38) into (37), with probability $1 - 2\delta$, we have: (1.) if $\theta = \frac{1}{2}$,

$$\|\nabla F(\mathbf{w}_{T+1}) - \nabla F_S(\mathbf{w}_{T+1})\|^2 = \mathcal{O}\left(\frac{\log(1/\delta)}{T} + \frac{\log^2(1/\delta)}{n^2} + \frac{F(\mathbf{x}^*)\log(1/\delta)}{n}\right); \qquad (39)$$

(2.) if $\theta \in (\frac{1}{2}, 1]$,

$$\|\nabla F(\mathbf{w}_{T+1}) - \nabla F_S(\mathbf{w}_{T+1})\|^2 = \mathcal{O}\left(\frac{\log^{(\theta+\frac{3}{2})}(\frac{1}{\delta})\log^{\frac{1}{2}} T}{T} + \frac{\log^2(1/\delta)}{n^2} + \frac{F(\mathbf{x}^*)\log(1/\delta)}{n}\right);$$
$$(40)$$

(3.) if $\theta > 1$,

$$\|\nabla F(\mathbf{w}_{T+1}) - \nabla F_S(\mathbf{w}_{T+1})\|^2$$

$$= \mathcal{O}\Big(\frac{\log^{(\theta+\frac{3}{2})}(\frac{1}{\delta})\log^{\frac{3(\theta-1)}{2}}(T/\delta)\log^{\frac{1}{2}}T}{T} + \frac{\log^2(1/\delta)}{n^2} + \frac{F(\mathbf{x}^*)\log(1/\delta)}{n}\Big). \qquad (41)$$

According to the Polyak-Łojasiewicz condition, we know

$$F(\mathbf{w}_{T+1}) - F(\mathbf{x}^*) \le \frac{1}{4\mu}\|\nabla F(\mathbf{w}_{T+1})\|^2$$

$$\le (2\mu)^{-1}(\|\nabla F(\mathbf{w}_{T+1}) - \nabla F_S(\mathbf{w}_{T+1})\|^2 + \|\nabla F_S(\mathbf{w}_{T+1})\|^2). \qquad (42)$$

Plugging the convergence bound in (38) and the generalization bound in (39)-(41) into (42), with probability $1 - 3\delta$, we have (1.) if $\theta = \frac{1}{2}$,

$$F(\mathbf{w}_{T+1}) - F(\mathbf{x}^*) = \mathcal{O}\Big(\frac{\log(1/\delta)}{T} + \frac{\log^2(1/\delta)}{n^2} + \frac{F(\mathbf{x}^*)\log(1/\delta)}{n}\Big);$$

(2.) if $\theta \in (\frac{1}{2}, 1]$,

$$F(\mathbf{w}_{T+1}) - F(\mathbf{x}^*) = \mathcal{O}\Big(\frac{\log^{(\theta+\frac{3}{2})}(\frac{1}{\delta})\log^{\frac{1}{2}}T}{T} + \frac{\log^2(1/\delta)}{n^2} + \frac{F(\mathbf{x}^*)\log(1/\delta)}{n}\Big);$$

(3.) if $\theta > 1$,

$$F(\mathbf{w}_{T+1}) - F(\mathbf{x}^*) = \mathcal{O}\Big(\frac{\log^{(\theta+\frac{3}{2})}(\frac{1}{\delta})\log^{\frac{3(\theta-1)}{2}}(T/\delta)\log^{\frac{1}{2}}T}{T} + \frac{\log^2(1/\delta)}{n^2} + \frac{F(\mathbf{x}^*)\log(1/\delta)}{n}\Big).$$

We choose $T \asymp n^2$, then we can get the following inequality with probability $1 - \delta$

$$F(\mathbf{w}_{T+1}) - F(\mathbf{x}^*) = \begin{cases} \mathcal{O}\left(\frac{\log^2(1/\delta)}{n^2} + \frac{F(\mathbf{x}^*)\log(1/\delta)}{n}\right) \text{ if } \theta = \frac{1}{2}, \\ \mathcal{O}\left(\frac{\log^{(\theta+\frac{3}{2})}(\frac{1}{\delta})\log^{\frac{1}{2}}n}{n^2} + \frac{F(\mathbf{x}^*)\log(1/\delta)}{n}\right) \text{ if } \theta \in (\frac{1}{2}, 1], \\ \mathcal{O}\left(\frac{\log^{\frac{3(\theta-1)}{2}}(n/\delta)\log^{(\theta+\frac{3}{2})}(\frac{1}{\delta})\log^{\frac{1}{2}}n}{n^2} + \frac{F(\mathbf{x}^*)\log(1/\delta)}{n}\right) \text{ if } \theta > 1. \end{cases}$$

The proof is complete. $\qquad \qquad \square$

