# OpenReview forum: "Sharper Bounds of Non-Convex Stochastic Gradient Descent with Momentum"
_ICLR.cc/2025/Conference — Submitted to ICLR 2025_

### Official Review · Reviewer_bCQy · 2024-11-03

**Soundness:** 2
**Presentation:** 3
**Contribution:** 2
**Rating:** 5
**Confidence:** 3

**Summary:**

This paper studies high probability bound, including both convergence bound (corresponding to ERM loss) and generalization bound (corresponding to population loss) of SGDM in the non-convex and smooth regime. Besides the general non-convex case, this paper also analyzes the scenario under PL condition and shows accelerated bounds with this additional assumption. The results work under a relaxed assumption of stochastic gradientsa, where the noises of stochastic gradients are generalized to the family of sub-Weibull distributions.

**Strengths:**

- This paper provides a systematic analysis of high probability bounds of SGDM under various scenarios.
- Regarding the technical aspect, this paper comes up with a uniformed analysis for different types of gradient noises, including sub-Gaussian, sub-Exponential, and heavy-tail noises, using the family of sub-Weibull distributions, which captures the degree of "heavy-tailness" by the parameter $\theta$. This is also reflected in all results which naturally implies that heavier tails leads to worse bounds.

**Weaknesses:**

I don't agree with authors' claim that this paper is the first one studying the generalization bound of SGDM and that the two referred papers, Li & Orabona 2020 and Cutkosky & Mehta 2021, provide convergence bound (defined w.r.t. the ERM loss $\nabla F_S$). I believe both referred papers actually provide generalization bound, e.g., see Li & Orabona Theorem 1 and Cutkosky & Mehta Theorem 2. Both theorems provide upper bounds on the generalization loss $\nabla F$ instead of the ERM loss $\nabla F_S$. Also I'd like to point out the difference in terms of data sampling: this paper considers sampling with replacement from a fixed dataset so that there could be repeated data, while the referred papers sample an i.i.d. data from the unknown population distribution in every iteration. This subtle difference is not mentioned in the paper. If my understanding is correct, the results of this paper is less novel than it claims.

As a related question, does Theorem 3.1 (and similarly Theorem 3.5) also hold for the generalization loss $\nabla F$? It seems to me that if we slightly modify Assumption 2.4 and 2.8 by replacing $\nabla F_S$ with $\nabla F$, then the same analysis in the proof in C.1 still holds: smoothness (eq. 5) is not affected; the martingale difference concentration on page 20 and the sub-Weibull concentration on page 21 still hold with $\nabla F_S$ replaced by $\nabla F$. In other words, is it true that the same analysis (under slightly different assumptions) provides a generalization bound?

As a disclaimer, I'm not an expert in the field of learning theory, and I could be wrong with my understandings. I will be glad to reevaluate the results if the authors point out I'm wrong.

**Questions:**

- See weakness for my main question.

- Line 1043: could the authors explain why $\xi_t$ is sub-Weilbull (or alternatively why we can apply Lemma B.4 on $\xi_t$)?

- eq. 1: miss brackets around $\exp$

---

### Official Review · Reviewer_sBpX · 2024-11-03

**Soundness:** 1
**Presentation:** 1
**Contribution:** 2
**Rating:** 3
**Confidence:** 3

**Summary:**

This paper provides high-probability convergence and generalization bounds for stochastic gradient descent with (heavyball) momentum (SGDM). Sharper bounds under PL assumption and Bernstein condition are also provided.

**Strengths:**

I believe SGDM is still used (although not really with decaying step size that this paper focuses on), so the provided theory for heavy-tailed noise can be relevant. The authors seem honest with their results, for instance explicitly stating the the provided results do not improve over standard SGD.

**Weaknesses:**

- The amount of improvement compared to related work seem very marginal. For instance, compared to Li & Orabona, a related work that this manuscript often refers to, the improvement is only from $\log(T/\delta)$ to $\log(1/\delta)$. I get that Theorem 3.1 also holds for heavy tailed noise ($\theta > 1$), but in that case the authors admit the bound does not improve over standard SGD.
- I am not an expert in generalization bound for stochastic gradient methods. Yet, a quick search shows [1], which is present in the references of this paper, but I could not find where it was cited. At least some comparison to [1] should be provided.
- SGDM with decaying step size is quite outdated at this point, with cosine or more sophisticated sche dules, for instance [2]. As such, I’m not sure how useful these results are.
- Experiments are shown for convex examples like logistic regression and Huber loss. What is the point of these experiments, when the entire paper is about non-convex bounds?
- Writing should be vastly improved. Many paragraphs and sentences are dense and long, hurting readability quite a bit: e.g., line 210 to 215 is one sentence, Remark 3.2, entire experimental setup from 466 to 492 is one paragraph… etc).

[1] Ramezani-Kebrya et al. (2024) “On the Generalization of Stochastic Gradient Descent with Momentum”

[2] Defazio et al. (2024) "The Road Less Scheduled”

**Questions:**

- $\tilde{O} (1/T^{1/2})$ is the same rate as Li & Orabona (2020)? What do you mean by “the convergence bounds are tighter than those of the related work”? (I’m aware that Li & Orabnoa assumes sub-gaussian noise).
- Assumption 2.8 needs to define quantities like $j_t$. I’m assuming this assumption has to hold for all $t$.
- Typo: line 292, “over SGD [1]”. What is ref [1]? Seems hard coded.

---

### Official Review · Reviewer_Cdqp · 2024-11-04

**Soundness:** 3
**Presentation:** 3
**Contribution:** 3
**Rating:** 6
**Confidence:** 3

**Summary:**

The paper studies Stochastic Gradient Descent with momentum (SGDM) and introduces theoretical convergence bounds and generalization bounds for SGDM. These bounds are tighter and faster than the theoretical results of related works in different settings.

**Strengths:**

For general convex case, the authors prove that the convergence bounds of SGDM are sharper than that of related works and present the first generalization bounds of SGDM. With the additional Polyak-Łojasiewicz condition, the convergence bounds of SGDM achieve a faster $O(1/T)$ rate.

**Weaknesses:**

Missing baselines for comparison in numerical experiments. (details in questions)

**Questions:**

Figures 1 and 2 exclusively present variations of SGDM. Including additional baselines would enhance the comparative analysis, providing insight into the performance of the proposed algorithm relative to established methods. These baselines could incorporate algorithms from previous literature on non-convex optimization.

---

### Official Review · Reviewer_2FSi · 2024-11-04

**Soundness:** 3
**Presentation:** 2
**Contribution:** 2
**Rating:** 5
**Confidence:** 2

**Summary:**

This paper derives high-probability convergence rates and generalization bounds for stochastic gradient descent with momentum (SGDM), a popular algorithm in practice. The bounds are first established for general non-convex functions, assuming sub-Weibull gradient noise. Then the authors move on to consider functions that satisfy the PL condition and derive improved bounds, with generalization bound independent of the dimension.

**Strengths:**

1. The authors provide novel convergence results for SCDM, which is a popular algorithm in practice but is much harder to analyze than SGD.

2. All the assumptions that this paper makes are followed by detailed discussions and comparisons to existing literature.

**Weaknesses:**

Although I'm not so familiar with related literature, it seems that there are many existing high-probability and generalization bounds established for various optimization methods. It is unclear what are the key differences (see also Questions)

**Questions:**

1. What do you think are the key challenges in obtaining these theoretical guarantees for SGDM? Since there are some existing results as indicated in Table 1 that are worse than the results in the current paper, I'm curious about what are the key technical improvements over their analysis.

2. By considering heavy-tailed gradient noise (large $\theta$), what additional algorithmic insights can we obtain for the analysis? e.g. how do you compare SGDM with other first-order stochastic optimization methods?

3. I'm not familar with the generalization literature; can you explain why you are considering $T=n/d$ in the general non-convex case and $T=n^2$ in the PL case?

---

### Official Review · Reviewer_UQJb · 2024-11-11

**Soundness:** 3
**Presentation:** 3
**Contribution:** 3
**Rating:** 6
**Confidence:** 3

**Summary:**

This paper presents high probability convergence rates and generalization bounds for SGD with momentum under a heavy tailed noise assumption for both general non-convex functions and under PL conditions.

**Strengths:**

The paper is well written and the results appear to be fairly general and novel to my knowledge, though I am not an expert in recent developments in this area.

**Weaknesses:**

The empirical section is left somewhat underbaked - it will be worthwhile presenting additional results on more realistic tasks/datasets/neural networks (including transformer style architectures).

**Questions:**

Can the authors present a clear comparison against vanilla SGD (without momentum) and how their results compare against existing results based on Algorithmic stability (for instance, Hardt et al 2016 that the paper cites)?

---

### Meta-Review · Area_Chair_4Xy7 · 2024-12-08

**Metareview:**

The authors do not respond to the reviewers' comments and questions while the overall assessment on this paper is negative.

This paper studies the high probability convergence bounds and generalization bounds of Stochastic gradient descent with momentum (SGDM). The authors claim their bounds are tighter than the related work. However, the improvement is by a logarithmic term. The authors also claim this is the first study on the generalization bound of SGDM, but Li & Orabona 2020 and Cutkosky & Mehta 2021 have obtained related results before. More discussion on the significance of this work compared to  Li & Orabona 2020 and Cutkosky & Mehta 2021 is needed.

**Additional Comments On Reviewer Discussion:**

The authors do not respond to the reviewers' comments and questions while the overall assessment on this paper is negative.

---

### Decision · Program_Chairs · 2025-01-22

Reject